*Method*

EMBO
Molecular Medicine

# Pre-analytical drivers of bias in bead-enriched plasma proteomics

Kathrin Korff [iD] [1], Johannes B Müller-Reif [iD] [1], Dorothea Fichtl[2], Vincent Albrecht [iD] [1], Alicia-Sophie Schebesta[1], Ericka C M Itang [iD] [1], Sebastian Virreira Winter[3], Lesca M Holdt[2], Daniel Teupser [iD] [2], Matthias Mann [iD] [1✉] & Philipp E Geyer [iD] [1,4✉]

## Abstract

**Bead-based enrichment is a promising strategy to improve depth in plasma proteomics by overcoming the dynamic range barrier. However, its robustness against pre-analytical variation has not been sufficiently characterized. Here, we systematically evaluate five plasma proteomics workflows, including three bead-based methods, a neat workflow, and a precipitation protocol using spike-ins of low-abundance proteins and defined cellular contaminants. We find that bead-based approaches enhance detection of low-abundance proteins but can be highly susceptible to systematic bias from platelet and PBMC contamination. This can inflate results by thousands of proteins, potentially explaining some of the high literature-reported numbers. A perchloric acid-based workflow shows resistance to erythrocyte and platelet-derived contamination. We investigate how centrifugation conditions, anticoagulant choice, and buffer-bead combinations modulate contamination profiles and demonstrate that bias can be mitigated by optimized sample handling. Altogether, we identify more than 13,000 different protein groups, including cellular components from the circulating proteome. Our results provide a quantitative framework for assessing workflow performance under variable sample quality and offer guidance for both biomarker discovery and quality control in clinical proteomics studies.**

**Keywords** Plasma Proteomics; Sample Quality; Bead-Based Enrichment; Biomarker Validation; Pre-Analytical Bias
**Subject Categories** Methods & Resources; Proteomics

## Introduction

Blood plasma is one of the most valuable and widely collected biofluids for clinical diagnostics, with protein-based laboratory tests constituting the largest proportion of routine clinical assessments (Anderson and Anderson, 2002; Geyer et al, 2017). The plasma proteome is of fundamental importance for monitoring health status, pathogenic processes and treatment (FDA-NIH Biomarker Working Group, 2016; Ignjatovic et al, 2019). Classical examples of protein biomarkers are cardiac troponins for myocardial infarction or liver enzymes like ASAT and ALAT for hepatic dysfunction, representing well-established diagnostic tests. Most of the currently used biomarkers were introduced decades ago, and there remains a significant need for the discovery of novel biomarkers for many diseases and treatments (Anderson et al, 2013; Deutsch et al, 2021).

The complexity of the plasma proteome poses tremendous analytical challenges for biomarker discovery. Only twenty proteins account for approximately 99% of the total protein content, creating an extraordinary dynamic range that spans more than twelve orders of magnitude (Anderson and Anderson, 2002; Lee et al, 2011). This dynamic range presents a fundamental challenge for comprehensive proteome analysis, particularly in detecting low-abundance proteins such as tissue leakage markers or signal molecules, which are important for disease understanding and have great potential as novel biomarkers (Li et al, 2024).

Mass spectrometry (MS)-based proteomics is a rapidly advancing technology that increasingly provides unbiased and hypothesis-free analysis of complex samples (Aebersold and Mann, 2016; Guo et al, 2025). Recent technological advances including data-independent acquisition (DIA) alongside fast and high-resolution mass analyzers have significantly boosted the achievable plasma proteome depth (Lancaster et al, 2024; Serrano et al, 2024).

As importantly, emerging sample processing techniques have expanded the range of quantifiable proteins by directly addressing the dynamic range issue. This includes bead-based techniques enriching a subset of the plasma proteome in a bead-corona (Blume et al, 2020; Wu et al, 2025) or precipitation-based approaches such as perchloric acid precipitation (Viode et al, 2023, 2024; Albrecht et al, 2025). These methods aim to reduce the dominance of high-abundance proteins while being sufficiently robust to maintain quantitative integrity of proteome-wide analyses, thereby enabling the exploration of low-abundance species.

While these technological improvements have enhanced plasma proteome coverage, successful biomarker discovery requires careful consideration of many factors beyond the number of identified plasma proteins. The proteomics community has established guidelines for biomarker development that address quality standards and emphasize proper cohort selection to ensure

[1]Department of Proteomics and Signal Transduction, Max Planck Institute of Biochemistry, Martinsried, Germany. [2]Institute of Laboratory Medicine, University Hospital, LMU Munich, Munich, Germany. [3]ions.bio GmbH, Martinsried, Germany. [4]Present address: ions.bio GmbH, Martinsried, Germany. ✉E-mail: mmann@biochem.mpg.de; geyer@ions.bio

statistical significance and clinical relevance (Hoofnagle et al, 2016; Mischak et al, 2010; Skates et al, 2013; Surinova et al, 2011). However, there remains a critical gap in systematically assessing proteome-wide effects of preanalytical sample handling. Considering that plasma samples are often collected during routine clinical practice with variable processing conditions, these pre-analytical factors are known to significantly impact study outcomes (Rai et al, 2005; Schrohl et al, 2008; Timms et al, 2007). While this variability affects single samples and patients, it is particularly problematic in case-control studies, where systematic differences in sample collection or processing between groups can lead to false biomarker candidates. Of particular concern are contaminations from blood cells, especially platelets and erythrocytes, which can introduce systematic bias in clinical studies. We previously reported that approximately half of the published plasma proteomics studies may be affected by such sample-related quality issues (Geyer et al, 2019).

In this study, we present a comprehensive evaluation of five distinct plasma proteomics workflows with regard to pre-analytic factors: a neat plasma workflow for simple and rapid analysis, a precipitation-based approach using perchloric acid with neutralization (PCA-N) (Albrecht et al, 2025), and three distinct bead-based enrichment methods using magnetic or non-magnetic beads with representative surface chemistries. Through carefully designed spike-in experiments using various blood cell types as well as the entire yeast proteome, we systematically assess the ability of each workflow to detect low-abundance proteins. We also characterize their susceptibility to common sample contaminants, including platelets, erythrocytes, and peripheral blood mononuclear cells (PBMCs). Additionally, we investigate how standard clinical centrifugation protocols and blood collection tubes affect plasma proteome analysis, revealing critical insights into the relationship between sample preparation parameters and proteome composition. We further ask if compromised samples can be 'rescued' by additional processing steps. Our findings shed light on the potential and limitations of bead-based workflows and offer practical guidance for both prospective study design and retrospective quality evaluation of archived samples.

## Results

### Study overview and investigated plasma proteomics workflows

We designed a comprehensive experimental framework to systematically evaluate bead-based enrichment in comparison to neat plasma and perchloric acid precipitation with neutralization (PCA-N). This involves experiments across multiple dimensions, including pre-analytical variations commonly encountered in clinical studies, examining the differential protein detection patterns across workflows and analyzing their quantitative performance (Fig. 1A–D).

We implemented five distinct plasma proteomics workflows, all automated on the Agilent Bravo Liquid Handling Platform to ensure reproducibility and high-throughput processing across 96 samples in parallel (Methods). On the LC-MS side, we employed a state-of-the-art workflow by coupling the Evosep chromatography system to the Orbitrap Astral (Stewart et al, 2023; Guzman et al, 2024). These workflows are meant to be representative of different analytical approaches with distinct trade-offs between throughput, depth, and susceptibility to sample quality issues. The neat plasma workflow serves as the baseline method, employing simple protein denaturation followed by enzymatic digestion, optimized for speed and simplicity. For deeper proteome coverage without beads, we evaluated perchloric acid precipitation with neutralization (PCA-N), which separates proteins based on their solubility and has shown resistance to certain contaminants. Two of our workflows utilize magnetic bead-based enrichment approaches, strong anion exchange (SAX) and Sera Sil-Mag 700 silica-coated superparamagnetic beads (Sera Sil 700) (Methods). Proteins bind to these magnetic beads, followed by on-bead denaturation and digestion, with peptides eluting during the digestion process. The fifth workflow uses non-magnetic beads with different binding characteristics and follows similar enrichment principles.

The figure also depicts our design for controlled spike-in experiments with key cellular contaminants, platelets, erythrocytes, and PBMCs, as well as the yeast proteome as a proxy for low-abundance proteins. This includes a range of analytical and pre-analytical conditions, different bead types and buffer systems, freeze-thaw cycles, centrifugation settings, blood collection tubes, and anticoagulants. This comprehensive setup enabled us to quantify each workflow's resilience to contamination and its ability to detect low-abundance proteins under clinically relevant scenarios (Dataset EV1).

To understand how each workflow alters the original plasma proteome composition by its characteristic selective enrichment and depletion patterns, we compared changes in abundance rank orders (Fig. 1E). Using the neat plasma workflow as our baseline, we ranked all proteins detected in neat plasma by their abundance and then tracked how each protein's signal intensity changes in the alternative workflows. This revealed that all workflows substantially reshape the plasma proteome by enriching many low-abundance proteins while depleting a subset of high-abundance proteins. For all of the methods, this bidirectional modulation alleviates the extreme dynamic range that typically challenges comprehensive plasma proteome analysis. The PCA-N workflow is characterized by a moderate reshaping of the plasma proteome, with consistent depletion across the abundance range, allowing detection of lower abundance species. In contrast, bead-based methods lead to more dramatic transformations. The SAX and Sera Sil 700 magnetic beads achieve substantial enhancement of previously low-signal proteins, while the non-magnetic bead workflow produces the most pronounced redistribution. These differential enrichment patterns directly determine which proteins can be reliably detected and quantified in plasma samples and provide guidance for workflow selection based on target protein characteristics and research objectives. We found that a small proportion of the low-abundance-bead-detected proteins overlap with neat plasma (<25% of the lowest 30% of the bead proteome), and these proteins are at the lowest-abundance edge of the neat plasma proteome (Appendix Fig. S1A). An enrichment analysis of bead-specific protein populations revealed tissue-specific markers, extracellular components, and proteins from various subcellular compartments. This indicates that bead-based methods preferentially capture biologically relevant secreted and tissue-derived proteins that circulate at low concentrations (Appendix Fig. S1B–D). Quantitative precision analysis across abundance ranges confirms that reduced precision for low-abundance proteins is a general feature of MS-based proteomics, affecting all workflow degrees (Appendix Fig. S1E).

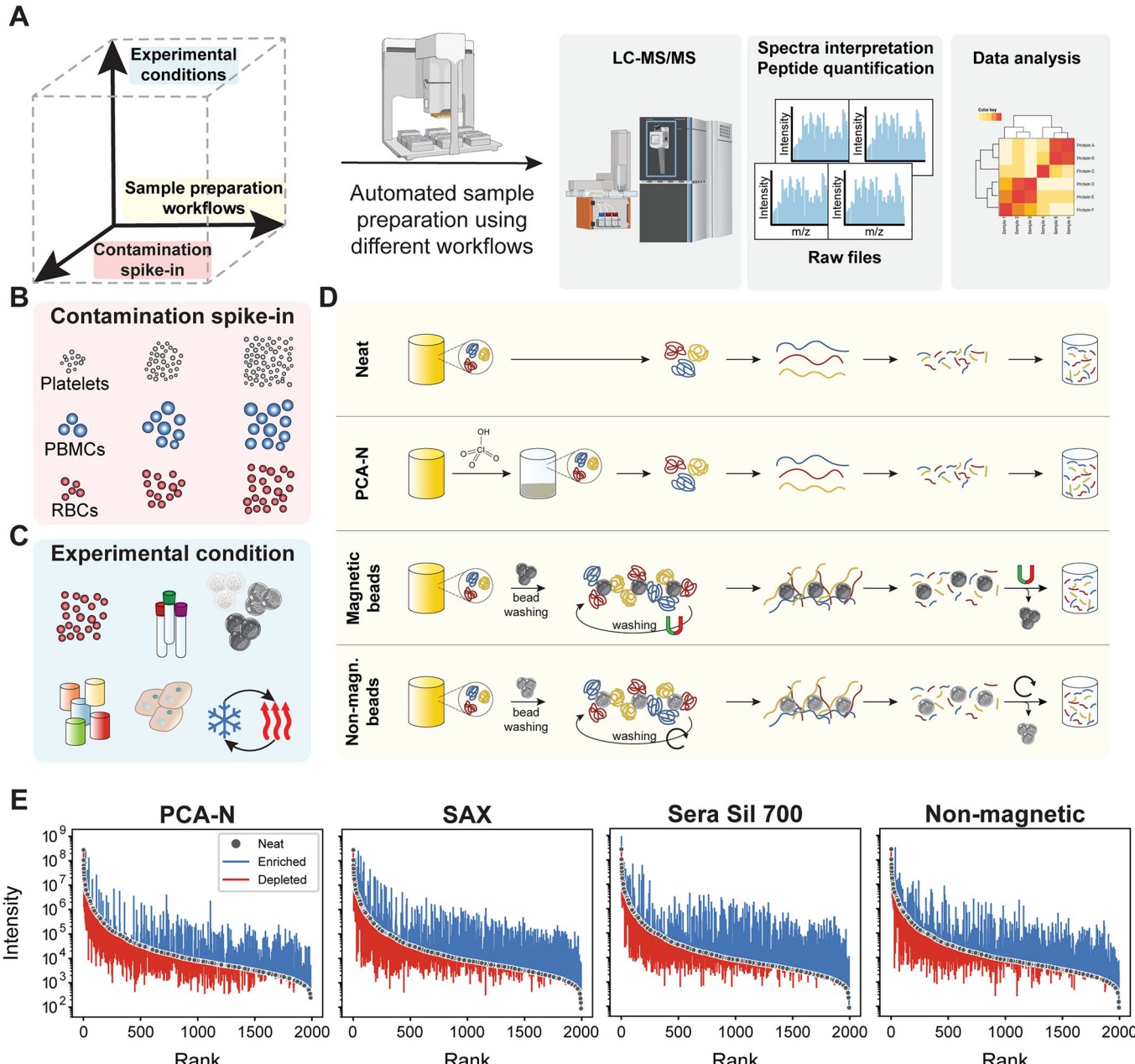

**Figure 1. Systematic experimental framework for the comparison of plasma proteomics workflows.**

(A) Three-dimensional study design exploring workflow variations, experimental conditions, and contamination spike-in experiments. (B) Spike-in experiments consisted of defined concentration steps of platelets, erythrocytes, and PBMCs. (C) Experimental conditions include different bead types, incubation buffer compositions, and pre-analytical variations such as freeze-thaw cycles. (D) Depiction of different workflows, including sample preparation of neat plasma, perchloric acid precipitation with neutralization, and bead-based methods using magnetic and non-magnetic beads, all processed with standardized downstream analysis. (E) Rank abundance plots comparing protein intensities across workflows, showing how each method reshapes the plasma proteome by enriching low-abundance proteins (blue) and depleting high-abundance proteins (red) relative to neat plasma preparation.

## Contamination analysis

Building on our previous efforts to systematically evaluate the role of cellular contamination in plasma samples (Geyer et al, 2019), we investigated how blood cells impacted plasma proteomics workflows. Next to erythrocytes (natural range 4–6 × $10^6$ cells/µL of blood; ~90 fL per cell) and platelets (1.5–4.5 × $10^5$ cells/µL, ~10 fL),

which are the most abundant blood cells, we added PBMCs (0.3–5 × $10^3$ cells/µL, ~250 fL) (Fig. 2). After isolating each cell type from whole blood, we verified their concentrations through complete blood counting (Fig. 2A; Dataset EV2). Additionally, we generated pure plasma with practically no cellular contamination. We then created precise contamination steps by adding defined cell numbers to ultra-pure plasma (Fig. 2B). These dilution series ranged from

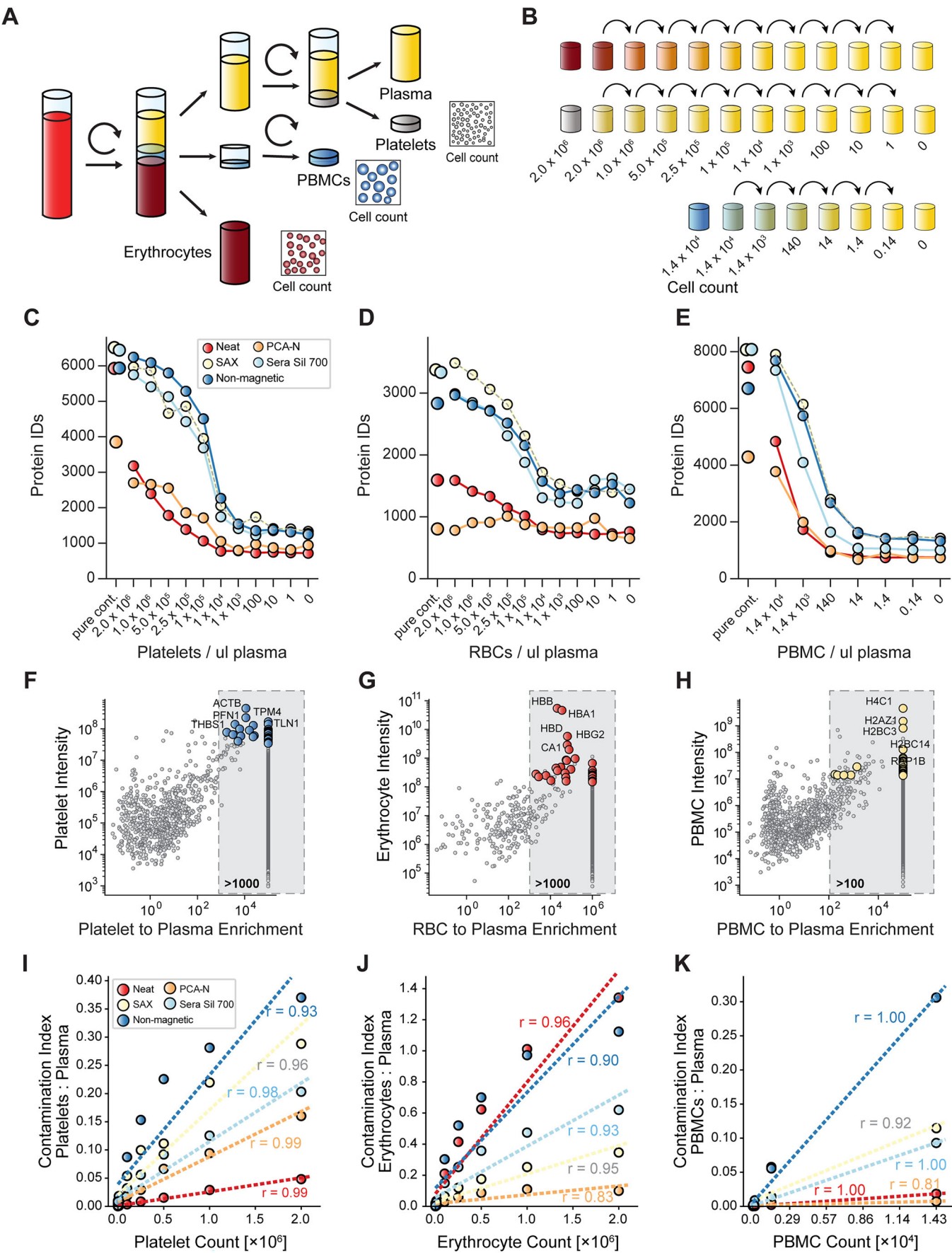

◄  **Figure 2.  Systematic analysis of cellular contamination effects on plasma proteomics workflows.**

(A) Experimental design for isolation of platelets, erythrocytes, and PBMCs from whole blood. (B) Contamination series design showing concentration gradients used for spike-in experiments (1 to $2.0 \times 10^6$ cells/µL for platelets and erythrocytes; 1 to $1.4 \times 10^4$ cells/µL for PBMCs). (C–E) Protein identifications across contamination series in five workflows plotted against cell concentrations for (C) platelets, (D) erythrocytes, and (E) PBMCs. Data shows the mean of four replicates. (F–H) Identification of cell-specific quality markers depicting protein intensity in pure contamination versus enrichment compared to plasma for (F) platelets, (G) erythrocytes, and (H) PBMCs. Selection criteria: >1000-fold enrichment (platelets/erythrocytes) or >100-fold (PBMCs); minimum of two peptides (precursors) per protein; high abundance (log10 intensity >7.5 for platelets, >8.2 for erythrocytes, >7.1 for PBMCs); good reproducibility (CV <20% for platelets/erythrocytes, <35% for PBMCs). Proteins only identified in the cellular proteome were aligned to the data with a slightly higher fold change. Top five markers are highlighted. These three panels are reused in Appendix Fig. S4A. (I–K) Contamination index versus cell count for (I) platelets, (J) erythrocytes, and (K) PBMCs across all workflows. Regression lines with correlation coefficients are shown.

single cells up to $2 \times 10^6$ cells. All samples were processed in quadruplicate for the five workflows, totaling 640 samples for analysis.

Analysis of protein identifications across contamination series revealed workflow-specific susceptibility patterns (Fig. 2C–E). Remarkably, when spiking 14,000 PBMCs per µL of plasma, we consistently identified more than 7000 proteins in the three bead protocols, not far from the >8000 proteins identified in pure PBMCs. Similarly, $2 \times 10^6$ platelets per µL plasma resulted in about 6000 platelet proteins, and $2 \times 10^6$ erythrocytes resulted in about 3000 proteins. Next, we investigated diminishing platelet concentrations, which revealed that bead-based workflows were exceptionally susceptible, with non-magnetic beads maintaining more than 4500 protein identifications even at moderate contamination of $1 \times 10^5$ platelets. This contrasted sharply with the neat workflow, where we identified 2100 proteins when spiking in $1 \times 10^6$ platelets/µL, roughly a doubling compared to no spike-in. PCA-N occupied a middle ground, at ~2600 proteins across high contamination levels but dropping to baseline at lower concentrations (Fig. 2C).

Erythrocyte contamination, however, produced fundamentally different patterns across workflows. PCA-N demonstrated notable resistance to erythrocyte contamination, maintaining nearly identical protein numbers (~800–900) regardless of spiked-in erythrocyte concentration, while all other workflows suffered strongly from erythrocyte contamination. Despite their lower absolute numbers in the contamination series, PBMCs produced the most dramatic effects per cell (Fig. 2E). All bead-based methods were highly susceptible to PBMC contamination down to 140 cells/µL, a full order of magnitude lower than for platelets and erythrocytes. This enhanced detection sensitivity likely reflected PBMCs' larger size and higher protein content compared to other blood cells. When examining combined contamination effects with intermediate levels (~500,000 erythrocytes/µL, 50,000 platelets/µL, and 500 PBMCs/µL), bead-based methods identified up to 8000 proteins compared to 3000 with the neat workflow. Collectively, these results demonstrate that blood cell contamination has the potential to substantially impact protein identification in a workflow-dependent manner, with bead-based methods showing high susceptibility to cellular protein contamination, while PCA-N shows particular resistance to erythrocyte contamination but remains susceptible to platelet contamination. Given this potential for misleading results, we next set out to systematically assess plasma quality in measured samples by establishing robust marker panels for each cell type that could serve as indicators of contamination (Geyer et al, 2019). We first compared protein abundances between pure contamination in neat plasma (Fig. 2F–H). To calculate fold-change ratios, we applied stringent filtering criteria, including >1000-fold enrichment for platelets and

erythrocytes and >100-fold for PBMCs, minimum precursor requirements, abundance thresholds, and reproducibility standards. This approach identified specific panels that included key proteins such as ACTB, PFN1, THBS1, TPM4, and TLN1 for platelets; HBB, HBA1, HBD, HBG2, and CA1 for erythrocytes; and H4C1, H2AZ1, H2BC3, H2BC14, and RAP1B for PBMCs. In characteristic distribution patterns, enrichment factors for platelet and erythrocyte markers exceeded $10^4$ (Fig. 2F,G; Dataset EV3). Compared to our previously published markers, 20 of the topmost stringent 30 proteins were identical, confirming consistency across different sample preparation workflows and different MS instruments, scan modes and software. This demonstrated the robustness of these cell-specific markers for quality assessment purposes (Appendix Fig. S2).

Next, we followed these quality panels across the various workflows using the dilution series experiment (Fig. EV1). Bead-based workflows again showed high susceptibility to platelet contamination compared to the neat workflow, whereas the PCA-N workflow showed reduced sensitivity to erythrocyte proteins. Again, PBMC markers were detectable at much lower cell counts (140 cells) compared to platelet and erythrocyte ones ($10^4$ cells).

To quantify the overall impact of cellular contamination on plasma proteomics workflows as a simple metric, we calculated a contamination index for each sample by dividing the summed MS intensity of the quality markers by the summed MS intensity of all other quantified proteins (Fig. 2I–K). For platelets, all workflows showed a strong correlation between cell count and contamination index, with distinctly different sensitivity profiles (Fig. 2I). As higher cell counts should directly result in higher contamination indices, fitting the data with a linear model worked well, with non-magnetic beads demonstrating the highest contamination index and steepest slope, followed by the magnetic bead-based workflows. The PCA-N workflow showed less substantial sensitivity to high platelet counts, while the neat workflow had a nearly flat slope, indicating minimal susceptibility to platelet contamination. This was in contrast to erythrocyte results, where the neat workflow showed the highest contamination index and steepest slope, followed by the non-magnetic beads (Fig. 2J). The magnetic bead workflows showed moderate sensitivity, while the PCA-N workflow was nearly unperturbed by erythrocyte contamination.

The contamination indices were approximately three times higher for erythrocytes than for platelets, partially due to some dominant proteins in the erythrocyte proteome and the relatively larger size of these cells. PBMC contamination (Fig. 2K) produced patterns similar to platelets, with non-magnetic beads showing the highest sensitivity, followed by magnetic beads, while both neat and PCA-N workflows showed minimal response. We found that bead-based workflows were better described by a power law model,

particularly for platelet and erythrocyte contamination in bead-based workflows, suggesting saturation effects at higher contamination levels across all cell types, presumably reflecting the finite binding capacity of the beads (Fig. EV2A–C).

To determine a possible disproportionate contribution to contamination by cellular components, we defined a cellular enrichment score as the ratio of the summed MS intensity of the top 30 cell-specific markers to the summed intensity of the top 30 plasma proteins. For platelet contamination, non-magnetic beads had high enrichment scores of 2.5 compared to less than 0.1 for the neat workflow (Appendix Fig. S3). The scores for SAX and Sera Sil 700 were 1.0 and 0.5, respectively. Erythrocyte enrichment scores were more uniform across workflows, with all methods, including neat, showing similar concentration-dependent patterns. For PBMCs, non-magnetic beads again showed the highest enrichment scores, while neat and PCA-N displayed minimal enrichment despite the disproportionate impact of these cells at low counts.

Additionally, we determined workflow-specific contamination markers by analyzing each cell type within each individual workflow separately (Appendix Fig. S4A; Dataset EV4). While these workflow-specific markers showed substantial differences in protein composition compared to neat plasma-derived markers (Appendix Fig. S4B–D), the resulting contamination indices were remarkably consistent across approaches (Appendix Fig. S4E–J).

Of note, using spike-in experiments with contamination levels that may occur in collected samples (e.g., $5 \times 10^5$ erythrocytes, $1 \times 10^4$ platelets, 1400 PBMCs), we identified ~8000 proteins with bead methods compared to 3000 with the neat workflow, while in worst-case high contamination scenarios, this number exceeded 10,000 proteins with bead-based methods.

These findings underscore our hypothesis that plasma contamination may distort proteome profiles in a workflow and cell type-specific manner. Bead-based methods demonstrate high susceptibility to cellular contamination, particularly from platelets and PBMCs, which can lead to dramatically inflated protein identification numbers that may not reflect the true circulating plasma proteome.

## Yeast spike-in as an external standard for quantitative workflow assessment

To complement the cellular contamination analysis, we next assessed each workflow's ability to detect low-abundance proteins. For this, we spiked-in the complete soluble yeast proteome as an external standard. This allowed us to systematically evaluate enrichment efficiency and dynamic range compression across workflows, without the complication of accessing cellular vs. plasma origin of measured proteins as is the case for human proteins.

Soluble yeast proteins obtained through cell disruption and centrifugation were spiked into plasma at 8 different ratios for the 5 workflows in quadruplicates (1:2, 1:4, 1:10, $1:10^2$, $1:10^3$, $1:10^4$, $1:10^5$, $1:10^6$, and pure plasma) (Methods) (Fig. 3A). To address the potential for residual intact yeast cells, the lysate was subjected to high-speed centrifugation at $16,000 \times g$ for 20 min after cell disruption. This rigorous centrifugation effectively pellets intact cells and large cellular debris, with careful supernatant collection to ensure that only soluble proteins were carried forward for subsequent analyses, minimizing contamination from residual intact yeast cells. In the pure yeast sample, we identified more than 4000 proteins over five orders of magnitude of MS signal in

our standard 100 SPD method, nearly the entire proteome (de Godoy et al, 2008) (Fig. 3C). This gradually declined as a function of dilution in plasma, with the number of identified yeast proteins a proxy for the ability of the workflow to detect low-abundance proteins. In fact, we observed substantial differences between workflows (Fig. 3B). For instance, in the 1:100 dilution ratio, the bead-based workflows demonstrated superior performance: Of the total detected 2286 yeast proteins, more than 70% (1676) were exclusive to bead-based workflows, with 44% of these consistently identified across all three methods. The non-magnetic bead workflow performed best, uniquely identifying 282 proteins (12% of total). In contrast, PCA-N and neat workflows were less effective, yielding 471 and 317 protein groups, respectively, with minimal unique identifications. As expected, the number of proteins commonly identified across all workflows decreased with increasing dilution, from 1128 at the 1:2 ratio to only 30 at 1:10,000, reflecting the expected concentration-dependent loss of detectable signal (Appendix Fig. S5).

We next evaluated the workflows for high, medium and low-abundance proteins (TDH3, RNQ1 and FAB1). The highly abundant protein TDH3 was detectable in the neat workflow down to $1:10^3$ dilution, while magnetic bead-based methods extended detection by an additional order of magnitude, and the non-magnetic bead workflow by two orders of magnitude (Fig. 3C). The PCA-N workflow maintained consistent median intensities but high variability across all dilution ratios. The bead workflows had high Pearson correlation coefficients of TDH3 intensity across yeast spike-in ratio ($r > 0.97$), although not for PCA-N. Five other highly abundant proteins confirmed this pattern (Appendix Fig. S6). The medium-abundant protein RNQ1 was detected down to 1:4 dilution in the neat workflow, which increased to $1:10^3$ with SAX and non-magnetic beads, showing superior correlation (Fig. 3C). The low-abundant protein FAB1 was not detectable in the neat workflow, but was found down to a 1:10 spike-in ratio in PCA-N and the bead-based workflows (Fig. 3C; Appendix Fig. S6).

For the top 100 most abundant yeast proteins, the neat workflow showed the expected quantitative behavior down to a 1% spike-in ratio (Fig. 3D). In the bead-based methods, intensity decreased less steeply than expected from dilution alone, enabling quantification down to 0.01% yeast in plasma, a 100-fold sensitivity improvement over the neat workflow. The observed fold-changes in PCA-N remained nearly constant across dilutions (Appendix Fig. S7).

To explore potential drivers of selective protein enrichment, we compared physicochemical properties of proteins enriched vs. depleted by each workflow using UniProt annotations. Six features were assessed: molecular weight, sequence length, isoelectric point, GRAVY hydrophobicity, aromaticity, and instability index. Proteins were considered strongly enriched or depleted if they differed by ≥80% from the neat workflow. Within each workflow, enriched vs. depleted proteins were compared using *t*-tests, and *p* values were visualized as $\log_{10}$-transformed values in a heatmap (Fig. 3E). Molecular weight and sequence length showed the strongest and most consistent effects: all bead workflows significantly enriched smaller, shorter proteins and depleted larger ones. Sera Sil 700 and non-magnetic beads also enriched proteins with significantly higher isoelectric points, suggesting a charge-driven enrichment bias. Hydrophobicity differences were modest, except in the PCA workflow, where hydrophilic proteins were significantly enriched. Aromaticity and instability index showed inconsistent trends across workflows.

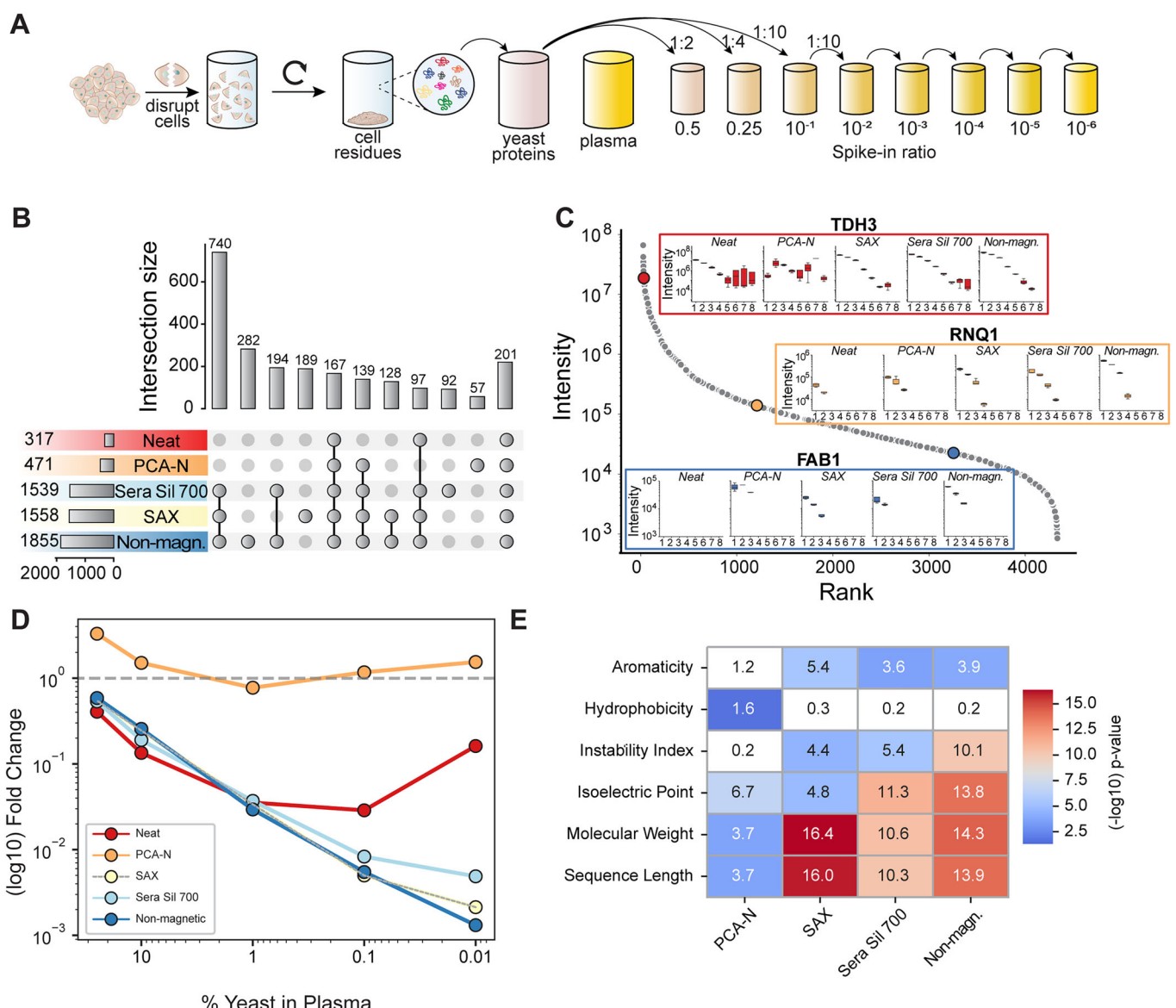

**Figure 3. Analysis of yeast protein spike-in across plasma proteomics workflows.**

(A) Experimental design for yeast protein preparation and plasma spike-in series. (B) Identified yeast proteins for a 1:100 ratio, visualized using an UpSet plot. (C) Dilution series of yeast proteins, presenting abundance-ranks of three representative proteins (TDH3, RNQ1, and FAB1). For this analysis, a DIA-NN search was performed with a 0.1% false discovery rate and data were filtered to include only proteins with at least three precursor peptides. The numbers 1–8 on the x-axis represent the dilution series with 1 = 1:2, 2 = 1:4, 3 = 1:10, 4 = $1:10^2$, 5 = $1:10^3$, 6 = $1:10^4$, 7 = $1:10^5$, and 8 = $1:10^6$ yeast:plasma ratio. (D) Signal intensities across decreasing yeast concentration. The percentage of yeast in plasma versus log10 fold change are shown from a 50% yeast reference sample for the top 100 yeast proteins across all workflows. (E) Heatmap of $-\log_{10} p$ values from $t$-tests comparing physicochemical properties between enriched and depleted proteins per workflow.

## Platelet activation

To shed more light on the pronounced sensitivity of bead-based protocols to platelet contamination, we designed an in-depth experiment with eight different types of beads versus seven buffer conditions. Surface chemistries ranged from negatively charged Sera Sil 700 (silanol hydroxyl groups) to positively charged SAX beads for strong anion exchange, Lewis acid-based $TiO_2$ and $ZrO_2$, chelating Ti-IMAC and Zr-IMAC NTA, and the non-magnetic beads. The binding milieus encompassed physiological conditions

(BTP/NaCl, HEPES, PBS), modified PBS variants (EDTA/PBS, PBSC), as well as conditions known to activate platelets ($CaCl_2$ and low pH environment of 2% TFA, pH 0.5). We tested these 56 conditions in quadruplicate on pure plasma containing platelets to generate a defined contaminated sample.

We first focused on total protein identifications in pure and platelet-contaminated plasma across all conditions, which varied between 650 and 2500 in pure and between 2000 and 3200 in platelet-contaminated plasma (Fig. 4A,B). Non-magnetic beads and HEPES buffer consistently yielded the highest protein

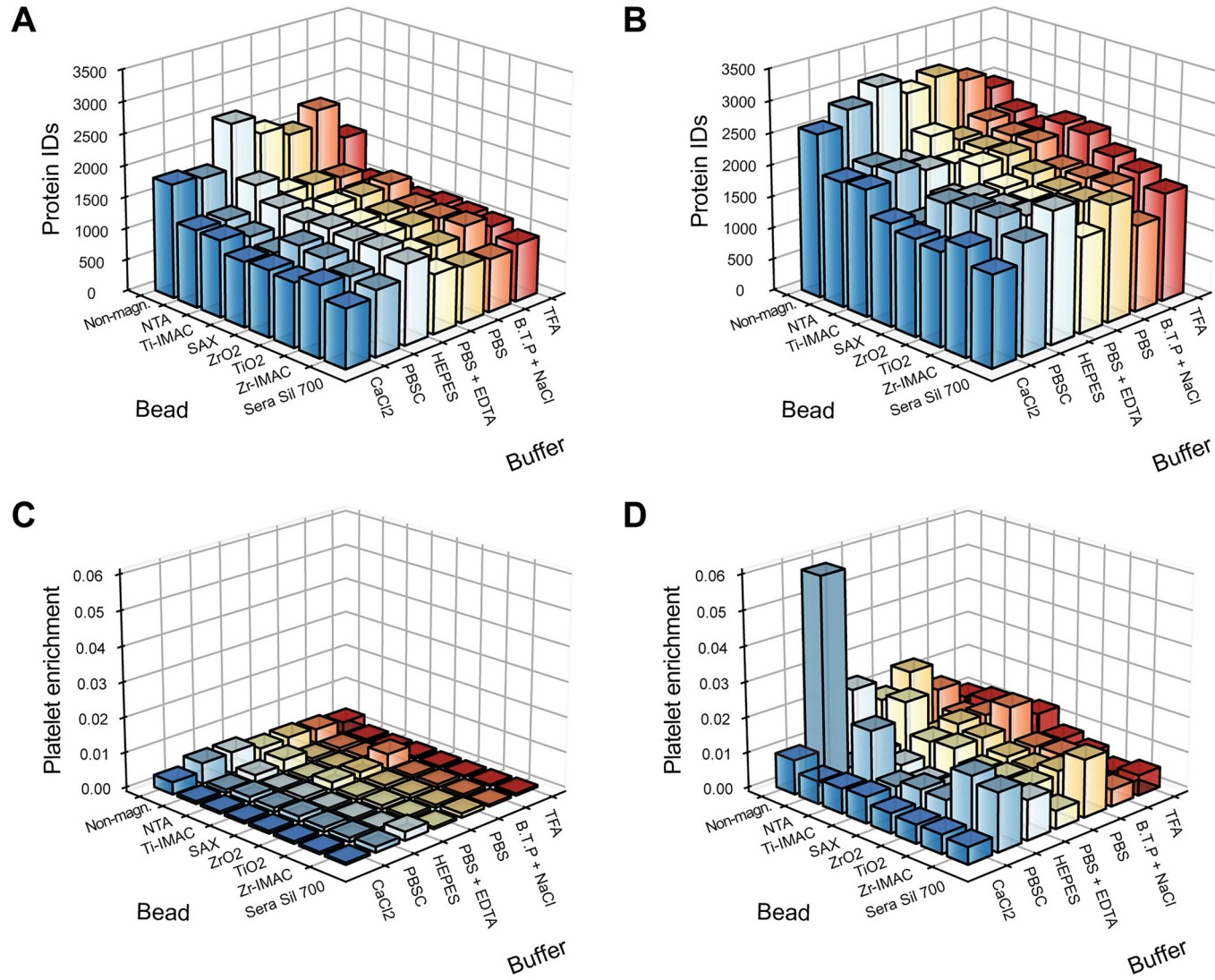

**Figure 4. Analysis of bead and buffer combinations in pure and platelet-contaminated plasma.**

(A) Number of protein identifications for bead-buffer combinations in pure plasma ($n = 4$). (B) Number of protein identifications in platelet-contaminated plasma ($n = 4$). (C) Platelet enrichment factor across bead-buffer combinations in pure plasma ($n = 4$). (D) Platelet enrichment factor across bead-buffer combinations in platelet-contaminated plasma ($n = 4$).

identifications in pure plasma. Trends generally remained similar in platelet-contaminated plasma, with notable exceptions such as generally high identifications when employing TFA or PBSC.

To elucidate mechanisms behind these variations in protein numbers, we calculated the platelet enrichment factor for each condition. As expected, pure plasma samples in general showed very low platelet enrichment compared to platelet-contaminated plasma (Fig. 4C,D). However, non-magnetic beads consistently had stronger platelet enrichment across all buffers. Note that non-magnetic bead workflows required an additional centrifugation step for bead washing, which may have co-pelleted residual platelets. This likely contributed to the higher protein numbers observed with non-magnetic beads, especially given our observation that even minimal contamination ($\sim 10^4$ platelets/µL) markedly affected this workflow. This interpretation was

supported by plasma with defined platelet spike-in, where non-magnetic beads and PBSC buffer resulted in high platelet enrichment compared to the other conditions (two to threefold when combined, Fig. 4D).

Physiologically, platelets are activated during injury, leading to protein secretion, but this process can also be triggered during plasma sample preparation. We evaluated the effects of different beads and buffers on platelet activation, focusing on five defined platelet markers (Appendix Fig. S8). Hierarchical clustering revealed that PF4V1 and PPBP had low intensities when employing $CaCl_2$, TFA buffers or Sera Sil 700 beads, which have a negative surface charge. These conditions activated platelets, leading to the release of these proteins (Singh et al, 2019). In contrast, non-magnetic beads with PBSC buffer led to high intensities for the activation markers, indicating platelet enrichment without activation.

## Rescue of platelet-contaminated samples

Given that already collected plasma samples can have significant platelet contamination, which is readily apparent in sensitive plasma proteomics workflows, especially bead-based ones, we asked to what extent this could be mitigated. To this end, we investigated defined platelet spike-ins, which we subjected to either direct processing or repeated freeze-thaw cycles (3x −80 °C for 15 min, thawing at 37 °C for 10 min). After splitting, these samples underwent centrifugation (3000 × g, 30 min) with careful supernatant collection, or no centrifugation. The resulting eight conditions were analyzed in quadruplicate using all five workflows (Fig. 5A).

The neat workflow was remarkably stable across all conditions with only minor variations in identifications and quantification (~800 ± 50 proteins, median CV ~ 13%), followed by PCA-N with a 35% increase in identifications. Identifications in magnetic bead-based workflows increased from 1400 proteins in pure plasma to 2700 proteins in platelet-contaminated samples, which decreased by only 15–20% after centrifugation (Fig. 5B). For non-magnetic beads, the reduction was larger, from 3700 to about 2000. Analysis of the CV revealed slight differences between freeze-thaw and direct processing conditions in several workflows, particularly in non-magnetic beads and SAX magnetic beads, although this effect was less pronounced in neat and PCA-N workflows (Fig. 5B).

A principal component analysis (PCA) pointed to platelet contamination as the dominant source of variation. While freeze-thaw cycles had less impact on overall proteome profiles compared to contamination, they notably increased data variability, particularly in bead-based workflows (Fig. 5C). In all cases except PCA-N, the PCA clearly separated centrifuged and non-centrifuged samples in the rescue experiments, supporting the benefit of this step. However, it should be noted that even after centrifugation, contaminated samples maintained distinct proteome profiles and did not fully return to the baseline of pure plasma samples. This was also apparent in violin plots showing Z-scored intensities of the top 100 platelet markers across all conditions (Fig. 5D). In pure plasma samples, intensities of these remained consistently low and stable across all processing conditions. In contrast, without centrifugation, platelet-contaminated samples retained high platelet marker intensities regardless of freeze-thawing. However, centrifugation still substantially reduced platelet marker intensity in contaminated samples. This reduction suggested improvement in sample quality, although samples remained distinct from pure plasma. This pattern was consistent across all workflows, with bead-based methods showing the most pronounced reduction in contamination, which was also evident at the single quality marker level (Fig. EV3A) and overall platelet panel (Fig. EV3B). To provide a comprehensive view of protein-level changes during contamination removal, we performed enrichment/depletion analysis comparing protein intensities before and after centrifugation across all workflows (Appendix Fig. S9). This analysis revealed workflow-dependent patterns where bead-based methods show simultaneous depletion of contaminating proteins and enrichment of low-abundance plasma constituents. This suggests that contamination removal may enhance the detection of genuine plasma proteins by freeing up bead binding capacity.

## Recommended plasma sample preparation conditions

While post-collection interventions such as centrifugation could reduce the impact of contamination, they did not universally restore sample quality. To identify optimal sample handling conditions, we systematically evaluated how pre-analytical variables, particularly centrifugation settings and tube types, affected contamination levels and downstream proteome profiles. We collected blood from 11 healthy individuals into standard EDTA tubes and EDTA gel separator tubes. Each tube was then centrifuged once using one of four different conditions: 500 × g for 7 min, 1000 × g for 7 min, 3000 × g for 7 min, and 3000 × g for 30 min. (Fig. 6A). Clinical blood counts revealed that gel tubes retained substantially more cellular material at low speeds, likely because the gel barrier activated only at higher g-forces. At 500 × g, gel tubes contained ~570,000 platelets per µL compared to 320,000 in standard tubes, a nearly twofold enrichment compared to whole blood. Erythrocyte contamination followed a similar trend, with higher levels in gel tubes under mild centrifugation. Increasing speed and time dramatically reduced contamination: platelet counts dropped below $6 \times 10^4$/µL at 3000 × g for 7 min and became undetectable after 30 min. Erythrocytes were effectively removed at 3000 × g in both tube types, and PBMCs decreased from ~670/µL in gel tubes at 500 × g to near-zero across higher-speed conditions (Dataset EV5). With these contamination profiles established, we processed all samples through our five distinct workflows and analyzed them by mass spectrometry. Protein identification numbers were strongly dependent on both centrifugation conditions and the chosen workflow (Fig. 6B). Bead-based enrichment identified substantially more proteins than neat or PCA-N at lower speeds. At 500 × g, all three bead workflows identified over 5500 proteins, whereas the neat and PCA-N workflows yielded only about 2000 proteins. The effect of increasing centrifugation was most dramatic for bead-based methods, where protein identifications dropped sharply with g-force. For instance, the number of identified proteins in the non-magnetic workflow decreased by ~70% between the lowest and highest centrifugation conditions. When comparing gel and non-gel tube types, they resulted in comparable protein identifications, with differences becoming apparent only at extended centrifugation times (Fig. 6C,D).

Principal component analysis highlighted centrifugation as the main factor shaping proteome profiles, with PC1 explaining between 26% (PCA-N) and 58% (SAX) of the variance (Fig. 6E). Bead-based workflows clearly clustered by g-force, especially at 3000 × g for 30 min. In contrast, PCA-N exhibited minimal separation, indicating low sensitivity to processing variation. The distinction between gel and no-gel tubes was most evident at lower speeds and nearly disappeared at 3000 × g for 30 min. Tube type influenced clustering mainly at low speeds.

Contamination indices provided a clear quantitative assessment of residual cellular content (Fig. 6F–H). Bead-based workflows showed markedly elevated platelet contamination indices at low centrifugation speeds, especially in gel tubes (Fig. 6F). Platelet contamination decreased steadily with increasing g-force and time, with over 90% reduction between the lowest and highest conditions. However, even at 3000 × g, bead workflows retained 10–20 times more platelet signal than the neat workflow. This shows that high centrifugation speeds are necessary to decrease platelet contamination, especially for bead-based workflows. The contamination bias of bead-based workflows completely settles in the range of the neat workflow only at long times. Nevertheless, even at very low concentrations, platelets are disproportionately captured and enriched by bead surfaces.

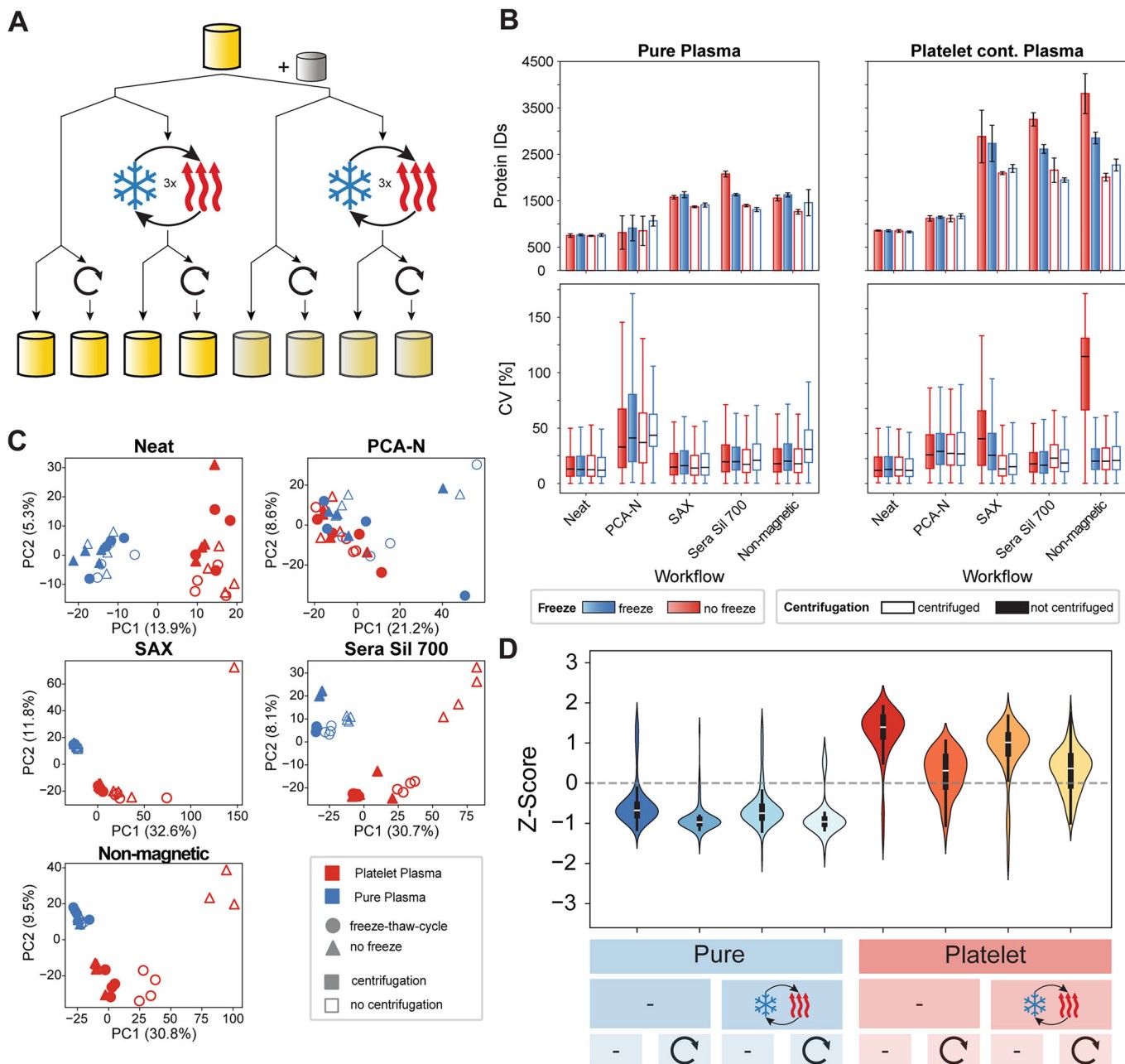

**Figure 5. Assessment of plasma processing strategies for platelet contamination rescue.**

(A) Experimental design for plasma sample processing evaluation. (B) Number of protein identifications (top) and CV distributions (bottom) for pure and platelet-contaminated plasma across five workflows. (C) Principal component analysis comparing different processing conditions. Percentage variance explained by PC1 shown in parentheses. One outlier (Platelet-contaminated plasma without freeze-thaw cycle and not centrifuged, replicate 1, Non-magnetic beads) was removed from the analysis, and K-nearest neighbors (KNN) imputation with three neighbors was applied to maintain dataset completeness ($n = 4$). (D) Violin plots showing Z-scored intensities of the top 100 platelet markers across all conditions and workflows. Blue represents pure plasma samples, red shows platelet-contaminated samples.

For erythrocyte and PBMC contaminations, neat and PCA-N showed low or no contamination, while bead-based enrichments were highly sensitive to both contaminations, especially at low centrifugation speed (Fig. 6G,H). PBMC contamination showed a less pronounced contamination pattern than platelets, with fivefold enrichment for bead-based methods over the neat plasma workflow.

To complement the aggregate contamination indices, we analyzed individual marker proteins across centrifugation conditions (Appendix Fig. S10B). PPBP, HBB, and H4C1 revealed workflow-specific enrichment patterns, underscoring the added value of single-protein analysis.

We next assessed the impact of anticoagulant type and gel separators by collecting blood into EDTA, Li-Heparin, and serum

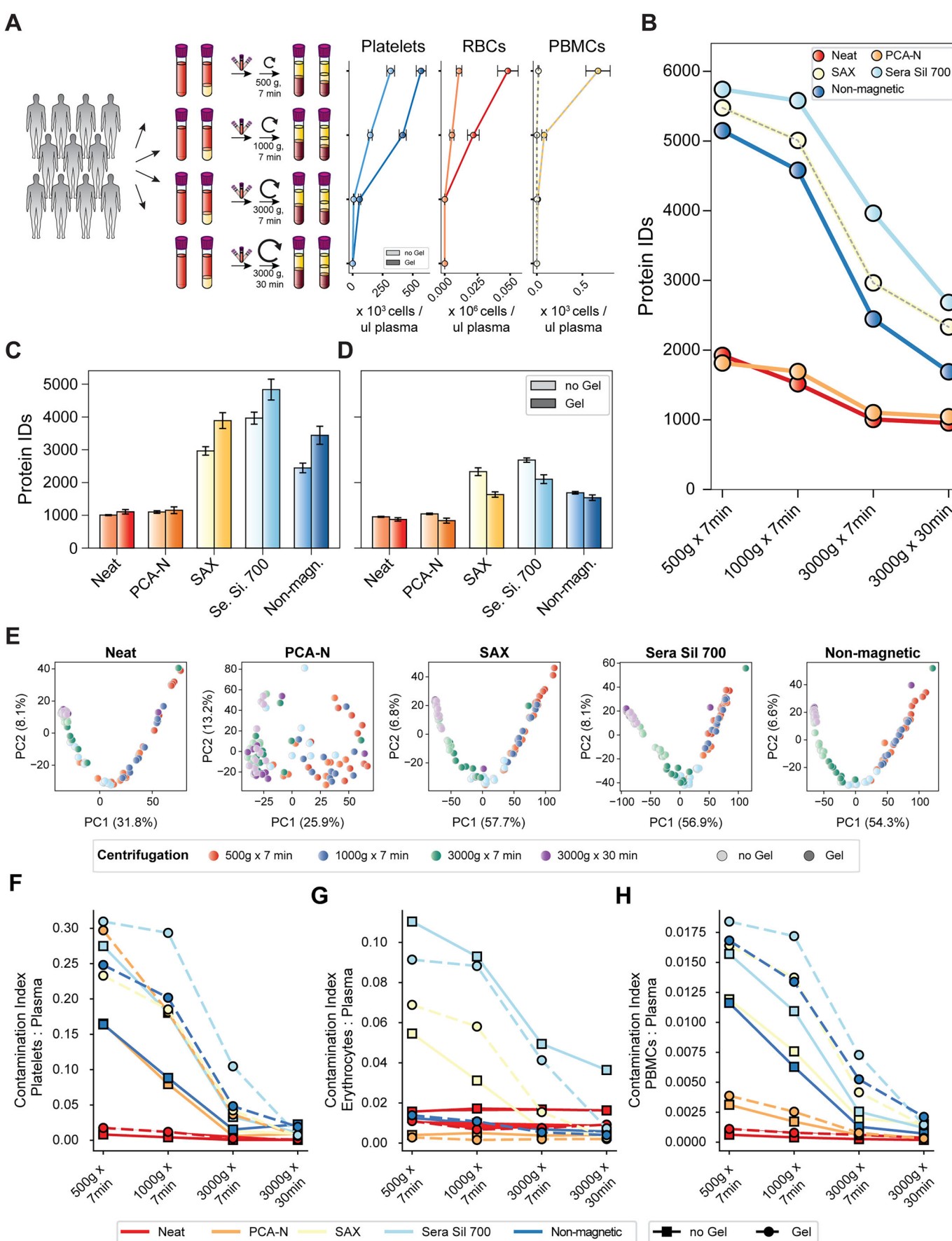

**Figure 6.  Impact of centrifugation conditions on plasma proteome analysis.**

(A) Experimental design: Blood was collected from 11 healthy individuals into standard EDTA tubes and EDTA gel separator tubes, followed by centrifugation at four different conditions (500 × $g$ for 7 min, 1000 × $g$ for 7 min, 3000 × $g$ for 7 min, and 3000 × $g$ for 30 min). Clinical measurements of cellular contamination (platelets, RBCs, PBMCs) were performed. (B) Number of protein identifications across five workflows plotted as lines, shown for standard EDTA tubes without gel separator ($n = 11$). (C) Number of protein identifications at 3000 × $g$ for 7 min comparing standard tubes versus gel separator tubes across all five workflows ($n = 11$). (D) Number of protein identifications at 3000 × $g$ for 30 min comparing standard tubes versus gel separator tubes across all five workflows ($n = 11$). (E) Principal component analysis of proteomics data for each workflow, colored by centrifugation condition and tube type. (F–H) Contamination indices plotted as lines across centrifugation conditions for (F) platelets, (G) erythrocytes, and (H) PBMCs. Solid lines represent standard tubes while dashed lines represent gel separator tubes, with different colors indicating different workflows.

tubes, with and without gels, from the 11 donors, all processed identically (3000 × $g$, 7 min; Fig. EV4A). Neat and PCA-N workflows were largely unaffected by matrix type (~1000 proteins identified), whereas bead-based workflows showed strong variation, with the highest identifications in EDTA plasma (up to 5000), followed by serum and Li-Heparin (Fig. EV4B). Principal component analysis showed clustering primarily by anticoagulant rather than gel use, most notably in the Sera Sil 700 workflow (Fig. EV4C). Platelet and erythrocyte contamination was highest in EDTA for bead workflows, while PCA-N remained resistant across conditions (Fig. EV4D).

Based on these findings, we recommend that the anticoagulant type should be standardized within a study. Combining matrices, such as EDTA and Li-Heparin-treated samples, should be strictly avoided. Although EDTA is a frequently used anticoagulant in biomarker research, our data suggest that other blood collection metrics like Li-Heparin and serum provide less contaminated samples for proteomics experiments, especially when applying bead-based workflows.

## Discussion

Our systematic comparison of five distinct plasma proteomics workflows, including three bead-based protocols (Blume et al, 2020; Wu et al, 2025), a perchloric acid-based precipitation neutralization (PCA-N) (Albrecht et al, 2025), and a conventional neat plasma workflow, reveals a critical trade-off between proteome depth and susceptibility to cellular contamination. Bead-based methods provide the highest protein identifications even in clean plasma but are acutely sensitive to contamination with platelets, erythrocytes, and PBMCs. In contrast, PCA-N shows strong resistance, especially to erythrocytes and, to a lesser extent, platelets. The neat workflow occupies a middle ground with moderate vulnerability but fewer identifications. The latter two technologies offer advantages of automation, low additional costs, and ease of use. Importantly, while these workflows differ in their degree of susceptibility, all plasma proteomics approaches are affected by cellular contamination to some extent, making quality monitoring and assessment of pre-analytical biases essential regardless of the chosen methodology (Table EV1).

Using soluble yeast proteins as an orthogonal spike-in strategy, we confirmed that bead-based methods quantitatively enrich low-abundance proteins beyond the capabilities of neat or PCA-N workflows. This enrichment was accurate across several orders of magnitude and confirmed dynamic range compression in bead-based approaches. PCA-N featured consistent protein detection

across abundance categories despite higher variability, suggesting selective protein depletion and enrichment mechanisms that enable low-abundance protein detection while resisting certain contamination types, a behavior also noted by Beimers et al (2025) in their comparison of acid-based depletion methods. Thus, the increased sensitivity is real, but comes at the cost of greater vulnerability to bias when working with non-ideal specimens. An additional limitation of bead-based enrichment is that proteins with complex post-translational modifications may bind to bead surfaces, potentially affecting both the discovery and quantitative analysis of diverse protein forms and introducing an additional layer of selectivity bias beyond cellular contamination effects.

This vulnerability has profound implications for biomarker discovery. In case-control studies, systematic differences in sample quality between groups can mimic or obscure true biological effects, particularly when using bead-based workflows. A small variation in platelet contamination may manifest as hundreds or thousands of additional proteins, compromising statistical power and potentially leading to spurious biomarker candidates. This calls into question the suitability of bead-enrichment methods for studies using archived or clinically collected plasma samples unless stringent quality control measures are in place. Our findings are particularly relevant for longitudinal studies where sample collection may span months or years with potential variations in handling protocols, and for cross-sectional and multicentric studies comparing different patient populations where sample collection conditions might systematically differ between sites.

To mitigate these risks, we had previously developed a three-step contamination control strategy that provides practical guidance for both prospective and retrospective studies: (1) assessing contamination in individual samples using our validated marker panels and excluding outliers relative to study-specific baselines, (2) detecting potential bias between study groups, and (3) correlating candidate biomarkers with cell-specific markers using global correlation maps to identify artifacts and distinguish genuine biological signals from technical effects. Building on our previous work (Geyer et al, 2019), we developed contamination indices and validated quality panels for platelets, erythrocytes, and PBMCs. These were consistent across workflows and instruments and enabled a reliable assessment of contamination at both sample and cohort levels. Notably, in the presence of contaminating cells, bead-based workflows result in an inflation of protein identification. While PBMCs occur at much lower concentrations than erythrocytes or platelets in blood, they produced substantial proteome alterations at just 140 cells/µL in bead-based workflows, highlighting the critical importance of controlling for cellular

contamination in studies examining immune-related conditions (Robinson et al, 2013; Messner et al, 2020).

Post-collection interventions, particularly centrifugation at 3000 × g for 30 min after thawing, can partially rescue compromised samples by reducing platelet marker signals, though not entirely eliminating them. This finding addresses a critical gap in current plasma proteomics guidelines, which typically recommend optimal collection procedures but provide limited guidance for handling already-compromised samples (Ignjatovic et al, 2019). We therefore recommend this centrifugation protocol as a standard step in any proteomic analysis of stored plasma.

Beyond rescue strategies, we systematically evaluated centrifugation conditions and anticoagulant types as pre-analytical variables. Low-speed centrifugation or the use of gel separator tubes were associated with high residual cellular content, particularly platelets. These effects were workflow-dependent: while neat and PCA-N workflows remained relatively unaffected, bead-based workflows were highly sensitive. Even after high-speed centrifugation, residual platelet signals remained detectable in bead-based methods, suggesting that platelets are selectively retained or enriched by certain bead surfaces

Our detailed examination of centrifugation conditions and tube types revealed critical insights for plasma sample preparation. At lower forces (500 × g, 1000 × g), gel separator tubes actually increased contamination compared to standard tubes, as the gel begins to move but does not form a complete barrier, creating turbulence (Bowen and Remaley, 2014; Sadgrove et al, 2024). Centrifugation at 3000 × g for 30 min provided the cleanest plasma across all tube types, minimizing contamination across all workflows. We therefore recommend this condition as a standard for plasma collection in proteomics studies, particularly those employing bead-based enrichments, although 3000 × g and 7 min may often be more practical. Consistency in tube type and anticoagulant (e.g., using EDTA plasma rather than serum or Li-Heparin) also proved critical. EDTA plasma yielded the highest and most stable identifications in bead workflows, whereas Li-Heparin introduced distinct proteome patterns, likely due to its effect on protein binding and coagulation. The complex relationship of pre-analytical variation and detectable proteome is emphasized throughout the plasma proteomics literature (Deutsch et al, 2021; Geyer et al, 2019; Ignjatovic et al, 2019).

Our findings are supported by a very recent preprint from Gao et al (2025). This thorough and well-designed study independently showed that nanoparticle-based enrichments are highly susceptible to blood cell contamination, particularly from platelets and erythrocytes. Their use of bovine plasma dilution series mirrors our yeast spike-in design and independently confirms the quantitative capability, but also the contamination risk of bead-based methods (Gao et al, 2025).

Taken together, our study provides both a cautionary note and a practical framework. Bead-based enrichments offer real and powerful gains in depth, but these must be weighed against a heightened risk of systematic bias. Our three-step contamination control strategy for biomarker studies provides practical guidance: Contamination assessment through calculating indices for each sample, bias detection by evaluating whether contamination markers differ between groups, and candidate validation by correlating biomarkers with cell-specific intensities. As MS instrumentation continues to improve, with platforms like the Orbitrap Astral pushing depth and throughput further (Stewart et al, 2023; Hendricks et al, 2024; Lancaster et al, 2024; Serrano et al, 2024), and new application areas of plasma proteomics expand (Bader et al, 2023; Niu et al, 2025), the importance of controlling for sample quality will only grow. As technological capabilities expand, there will be a growing need to refine and broaden contamination panels to include additional cell types and to further resolve PBMCs into their subpopulations. With the ongoing advancements in instrumentation and novel strategies to increase the dynamic range of the plasma proteome, challenges such as sample contamination can still arise. However, the MS-based proteomics field now has powerful strategies to counter these negative effects. These developments greatly increase the likelihood of discovering true biomarker candidates that are worthwhile to carry forward to clinical validation.

In conclusion, our study underscores that sample quality, particularly in the form of cellular contamination, remains a key determinant of plasma proteomics data. By identifying the strengths and vulnerabilities of commonly used workflows, we offer guidance for selecting and optimizing methods based on sample context. We encourage researchers to apply rigorous contamination assessment and to match analytical depth with robustness, thereby ensuring that plasma proteomics fulfills its promise in biomarker discovery and translational research.

# Methods

**Reagents and tools table**

| Reagent/resource | Reference/source | Identifier/catalog No. |
|---|---|---|
| **Chemicals, enzymes, and other reagents** | | |
| Optima® LC/MS-grade Formic acid (FA) | Fisher Chemical | A117-50 |
| Optima® LC/MS-grade Acetonitrile (ACN) | Fisher Chemical | A955-212 |
| Optima® LC/MS-grade water | Fisher Chemical | W6-4 |
| Sodium Chloride | Sigma-Aldrich | S7653 |
| HEPES PUFFERAN® ≥99, 5%, Buffer Grade | Carl Roth | HN78.3 |
| Trifluoroacetic acid for synthesis | Sigma-Aldrich | 8.08260 |
| CHAPS, 98+% | Thermo Scientific Chemicals | B21927.06 |
| BIS-TRIS-Propan, ≥99.0% (titration) | Sigma-Aldrich | B6755 |
| Triethylammonium bicarbonate (TEAB) buffer, 1.0 M, pH 8.5 ± 0.1 | Sigma-Aldrich | T7408 |
| Tris(2-carboxyethyl)phosphine hydrochloride (TCEP-HCl) | Thermo Fisher Scientific | 20491 |
| Chloroacetamide (CAA) | Sigma-Aldrich | C0267 |
| n-Dodecyl β-D-Maltoside (DDM) | Sigma-Aldrich | D4641 |
| Dithiothreitol (DTT) | Sigma-Aldrich | D0632 |
| Trizma® pre-set crystals, pH 8.5 | Sigma-Aldrich | T8818 |
| Trypsin, proteomics grade | Sigma-Aldrich | T6567 |
| Lysyl Endopeptidase, Mass spectrometry Grade (LysC) | FUJIFILM Wako | 125-05061 |

| Reagent/resource | Reference/source | Identifier/catalog No. |
|---|---|---|
| MagReSyn® SAX beads | ReSyn Biosciences | MR-SAX005 |
| MagReSyn® NTA beads | ReSyn Biosciences | MR-NTA002 |
| MagReSyn® TiO₂ beads | ReSyn Biosciences | MR-TID002 |
| MagReSyn® ZrO₂ beads | ReSyn Biosciences | MR-ZRD002 |
| MagReSyn® Ti-IMAC beads | ReSyn Biosciences | MR-TIM002 |
| MagReSyn® Zr-IMAC beads | ReSyn Biosciences | MR-ZRM002 |
| Sera Sil-Mag 700 silica-coated superparamagnetic beads | Merck | GE29357373 |
| OmniProt™ | Westlake Omics | N/A |
| **Software** | | |
| DIA-NN | https://github.com/vdemichev/DiaNN | version 1.8.1 |
| Python | https://python.org | Version 3.12.10 |
| **Others** | | |
| twin.tec® PCR Plate 96-well LoBind | Eppendorf | 0030129512 |
| twin.tec® PCR Plate 384-well LoBind | Eppendorf | 0030129547 |
| Deepwell Plate 96-well, 1000 µL | Eppendorf | 951032603 |
| ThermoMixer® C | Eppendorf | 5382000015 |
| Mastercycler™ X50h | Eppendorf | 6316000019 |
| Heat Sealer S200 | Eppendorf | 5392000030 |
| DynaMag™-96 Side Skirted Magnet | Thermo Fisher Scientific | 12027 |
| Nanodrop 2000 Spectrophotometer | Thermo Fisher Scientific | ND-2000 |
| Bravo robot | Agilent | N/A |

## Methods and protocols

### Blood collection and fractionation

If not otherwise indicated, whole blood was collected by venipuncture into EDTA-containing tubes (9 mL). The study was approved by the Ethics Committee of LMU Munich (Reg. No. 17-012). Informed consent was obtained from all human subjects and confirmed that the experiments conformed to the principles set out in the WMA Declaration of Helsinki and the Department of Health and Human Services Belmont Report.

For the isolation of blood components, the tubes were first centrifuged at $500 \times g$ for 7 min to separate platelet-rich plasma from the cellular components. The supernatant (platelet-rich plasma) was carefully transferred to a new centrifugation tube and centrifuged again at $500 \times g$ for 7 min. This supernatant was collected and centrifuged at $3000 \times g$ for 7 min. After this step, the supernatant was subjected to a final centrifugation at $3000 \times g$ for 7 min to obtain platelet-free plasma, with care taken to avoid the bottom ~500 µL to prevent platelet contamination. For platelet isolation, the pellet from the $3000 \times g$ centrifugation was resuspended in 4 mL PBS/EDTA (1.6 mg/mL EDTA), centrifuged at $3000 \times g$ for 7 min, and the supernatant was discarded. This washing step was repeated once more, resulting in purified

platelets. To isolate erythrocytes, the pellet from the initial $500 \times g$ centrifugation was transferred to a 15 mL tube and centrifuged at $3000 \times g$ for 7 min. The supernatant, buffy coat, and top 1 mL of erythrocytes were discarded. The remaining erythrocytes were washed twice by resuspending in 4 mL PBS/EDTA and centrifuging at $3000 \times g$ for 7 min, with removal of the supernatant and top layer of cells after each wash, yielding purified erythrocytes.

PBMCs were isolated from whole blood collected in CPT-Vacutainers. The vacutainers were gently inverted ten times immediately after collection and centrifuged at $1650 \times g$ for 20 min. After centrifugation, the content of the CPT-Vacutainers was transferred to 50 mL tubes. The tubes were filled with PBS buffer up to 50 mL and gently inverted to mix. These samples were then centrifuged at $400 \times g$ for 15 min. The supernatant was carefully removed, and the cell pellet was retained. A filter was placed on a second 50 mL Falcon tube, and 10 mL of PBS buffer was added to the cell pellet. The cells were carefully resuspended and filtered into the second tube. Additional PBS buffer was added through the filter to reach the 50 mL mark. This suspension was centrifuged at $300 \times g$ for 15 min. After discarding the supernatant, the cell pellet was resuspended in 1600 µL of PBS buffer.

Cell counts were determined using a Sysmex XN 1000/9100 automated hematology analyzer (Sysmex Corporation) according to standardized laboratory procedures. This platform employs multiple measurement technologies optimized for specific cell types to ensure accurate quantification across diverse sample conditions. Platelet counts were assessed using a dual-methodology approach. The primary measurement utilized impedance measurement with hydrodynamic focusing, where platelets passing through a microaperture generate electrical pulses proportional to their volume. For samples with potential counting interferences or suspected thrombocytopenia, the analyzer additionally employed fluorescence flow cytometry (PLT-F), which labels platelets with a proprietary fluorescent marker for enhanced detection and discrimination from other cellular elements. Erythrocyte enumeration was similarly performed via impedance measurement with hydrodynamic focusing, with each erythrocyte generating a voltage pulse proportional to its volume as it passes through an electrically charged aperture. Total leukocyte counts were determined using fluorescence flow cytometry. This technique employs fluorescent dyes that differentially stain cellular components, allowing the analyzer to distinguish leukocytes based on their size, internal complexity, and fluorescence characteristics, providing accurate quantification of white blood cells in the samples.

### Yeast protein preparation

Yeast proteins for spike-in experiments were prepared from the *Saccharomyces cerevisiae* strain BY4741. Briefly, 2 mg of yeast was mixed with 200 µL of PBS and subjected to bead milling to disrupt the cells and extract yeast proteins. To address the potential for residual intact yeast cells, the lysate was subjected to high-speed centrifugation at $16,000 \times g$ for 20 min. This rigorous centrifugation effectively pellets intact cells and large cellular debris, which are excluded from the resulting supernatant that contains the extracted yeast proteins. The supernatant was carefully collected without disturbing the pellet and used for generating the yeast dilution series in plasma samples.

### Sample preparation workflows

All sample preparation workflows were automated using an Agilent Bravo Liquid Handling Platform (Geyer et al, 2016).

Neat Plasma Workflow: Plasma (1 µL) was combined with 50 µL lysis buffer (100 mM Tris pH 8.0, 40 mM chloroacetamide, 10 mM TCEP). Samples were heated at 95 °C with agitation for 10 min on a Thermomixer C. After cooling to room temperature, 10 µL digestion buffer (8 µL lysis buffer, 1 µL trypsin [0.5 µg/µL], 1 µL LysC [0.5 µg/µL]) was added, and proteins were digested at 37 °C for 16 h. The digestion was terminated by the addition of an equal volume of 0.2% TFA.

Perchloric Acid precipitation with neutralization (PCA-N): Plasma (5 µL) was diluted in 20 µL ddH$_2$O, followed by the addition of 25 µL 1 M perchloric acid. Samples were agitated at 4 °C for 1 h, followed by centrifugation at 4000 × $g$ for 20 min at 4 °C. The supernatant (24 µL) was collected and combined with 8 µL 1.4 M sodium hydroxide solution to adjust pH to 8–8.5. Lysis buffer (8 µL; containing 40 mM chloroacetamide, 20 mM DTT, 0.01% DDM, 60 mM TEAB) was added. Proteins were digested using trypsin/LysC (1.6 µL: 0.4 µL trypsin [0.5 µg/µL] and LysC [0.5 µg/µL each, 10.5 µL 1 M TEAB, 0.7 µL water) and digestion was stopped with TFA (final concentration 0.5%) (Albrecht et al, 2025). The peptide mixtures were analyzed by LC-MS/MS.

Magnetic bead-based enrichment (SAX, Sera Sil 700, TiO$_2$, ZrO$_2$, Ti-IMAC, Zr-IMAC, NTA): Magnetic beads (10 µL slurry, 20 mg/mL, MagReSyn® SAX, MagReSyn® TiO$_2$, MagReSyn® ZrO$_2$, MagReSyn® Ti-IMAC, MagReSyn® Zr-IMAC, MagReSyn® NTA, Resyn Biosciences, Sera Sil 700: Sera Sil-Mag 700 silica-coated superparamagnetic beads) were washed three times with incubation buffer (SAX: 100 mM Bis-Tris Propane, pH 6.3, 150 mM NaCl; Sera Sil 700: 1 M HEPES) using magnetic separation (Wu et al, 2025). Plasma (10 µL) was combined with 100 µL incubation buffer (SAX: 50 mM Bis-Tris Propane, pH 6.5, 150 mM NaCl; Sera Sil 700: 1 M HEPES) and prewashed beads, followed by incubation at 37 °C for 30 min with agitation. After magnetic separation, bead-bound proteins were washed with washing buffer. Proteins were denatured, reduced, and alkylated while bound to the beads by the addition of lysis buffer (50 µL, 100 mM Tris pH 8.0, 40 mM chloroacetamide, 10 mM TCEP) followed by heating at 95 °C for 10 min. Digestion proceeded as described for the neat workflow, resulting in peptide elution from the beads. The eluted peptides were acidified and analyzed by LC-MS/MS.

Non-magnetic bead workflow: The non-magnetic bead workflow utilized non-magnetic beads (OmniProt™, Westlake Omics), requiring centrifugation at 4000 × $g$ for 20 min for all washing and collection steps. Beads were resuspended and washed in PBS. Plasma protein binding was performed in PBS containing 0.05% CHAPS, followed by washing of bead-bound proteins with 33% PBS. Proteins were denatured, reduced, and alkylated while bound to the beads by adding lysis buffer (33.3 µL; 40 mM chloroacetamide, 20 mM DTT, 0.01% DDM, 60 mM TEAB). Digestion proceeded as described for the neat workflow, resulting in peptide elution from the beads. The eluted peptides were acidified and analyzed by LC-MS/MS.

### Data acquisition by mass spectrometry

Samples were analyzed using the Evosep One liquid chromatography system (Evosep) (Bache et al, 2018) coupled to an Orbitrap Astral mass spectrometer (Stewart et al, 2023; Hendricks et al, 2024) (Thermo Fisher

Scientific). Chromatographic separation was performed on an 8 cm Aurora Rapid XT UHPLC column (AUR3-80150C18-XT, Ionopticks) at 50 °C using the "100 samples per day" method with pre-formed gradients and a total runtime of 11.5 min per sample. Mobile phases consisted of 0.1% formic acid in water (buffer A) and 0.1% formic acid in acetonitrile (buffer B). For each sample, 200 ng of peptides were loaded onto C-18 tips (Evotip Pure, Evosep) according to the manufacturer's protocol. Mass spectrometric analysis was performed using a data-independent acquisition (DIA) method (Guzman et al, 2024). The ion source was operated with a static spray voltage of 1900 V in positive ion mode, and the ion transfer tube temperature was set to 280 °C. FAIMS (high-field asymmetric waveform ion mobility spectrometry) was utilized in standard resolution mode with a compensation voltage (CV) of −40 V and a carrier gas flow of 3.5 L/min. Full MS1 scans were acquired in the Orbitrap analyzer at a resolution of 120,000 FWHM over a scan range of 380–980 m/z. The RF lens was set to 40%, and a normalized AGC target of 500% with a maximum injection time of 3 ms was used. DIA MS2 scans covered the same mass range (380–980 m/z) divided into 150 isolation windows of 4 Th each with window placement optimization enabled. MS2 spectra were acquired with an HCD collision energy of 25%, a normalized AGC target of 500%, and a maximum injection time of 7 ms. The scan range for fragment ions was set to 150–2000 m/z. Data were collected in profile mode for MS1 and centroid mode for MS2 scans. The expected chromatographic peak width was set to 5 s, and advanced peak determination was enabled to optimize the duty cycle. Quality control samples were analyzed regularly throughout the analytical sequence to monitor system performance and stability.

### Data analysis/spectral search

Each raw file was converted to the mzML format using Thermo Fisher Scientific's ThermoRawFileParser (v4.3) with format parameter set to 1, metadata parameter set to 0. The conversion was processed in parallel on a high-performance computing. The resulting mzML files were processed using DIA-NN (version 1.8.1) (Demichev et al, 2020) on a high-performance computing cluster. The searches were performed against a human UniProt Swiss-Prot isoform database that included oxidation and N-terminal acetylation modifications using the DIA-NN built-in in silico library prediction. All files were analyzed with match-between-runs enabled using the "--use-quant" and "--reanalyse" parameters. The following parameters were applied: peak centering, smart profiling, retention time profiling, and relaxed protein inference. Further, mass accuracy was set to 10 ppm for both MS1 and MS2 scans and the scan window to 7. False discovery rate was controlled at 1% at the peptide-to-spectrum match level. Protein quantification was performed using MS1 and MS2 data with no interference signal removal. Quantitative matrices were generated for downstream statistical analyses.

### Bioinformatics analysis

All bioinformatic analyses were performed using Python in Jupyter notebooks. Data processing was conducted using pandas for manipulation of proteomics data matrices and NumPy for numerical operations, while visualization and statistical analyses were employed using scipy, matplotlib, and seaborn packages. For protein quantification, we used the protein-level reports from DIA-NN output. Protein intensities were log10-transformed prior to further analysis. Where complete data matrices were required, e.g.,

### The paper explained

#### Problem

Blood plasma is widely used for biomarker discovery and diagnostics, but its high complexity and variability in sample quality present major challenges. Bead-based enrichment workflows offer improved detection of low-abundance proteins in MS-based proteomics. However, their susceptibility to pre-analytical variation, such as contamination from blood cells, is not well characterized. Thus, bead-based workflows may greatly inflate protein counts and introduce systematic bias, particularly in clinical studies using archived or variably processed samples.

#### Results

We compared five plasma proteomics workflows, encompassing three bead-based methods, neat plasma, and perchloric acid precipitation, using spike-ins of blood cell contaminants and a yeast proteome standard. Bead-based methods enabled deeper proteome coverage but were highly sensitive to platelet and PBMC contamination. This persisted even at low levels, often adding thousands of proteins to a sample's proteome. Centrifugation force, tube type, and anticoagulant choice significantly influenced contamination levels. We developed reproducible marker panels and contamination indices to assess and compare sample quality across workflows and conditions.

#### Impact

Our findings that pre-analytical factors and workflow choice can greatly influence plasma proteome data provide a cautionary antidote to the "numbers game" in plasma proteomics. Our marker panels and indices provide practical tools for identifying compromised samples and improving data interpretation in biomarker studies. This work supports more informed method selection and promotes better standardization in clinical proteomics, particularly when applying bead-based enrichment in variable sample settings.

for principal component analysis, K-nearest neighbors imputation (n neighbors = 3) was applied to handle missing values.

### Contamination analysis

We calculated contamination indices by dividing the summed intensity of cell-specific marker proteins by the summed intensity of all other quantified proteins. The relationship between cell counts and contamination indices was evaluated using both linear and non-linear regression models. For evaluation of cellular contamination, enrichment scores were calculated by dividing the summed intensity of the 30 defined cell-specific marker proteins by the summed intensity of the top 30 most abundant plasma proteins, to highlight the relative abundance of contamination markers compared to the dominant plasma proteome components.

## Data availability

The proteomics datasets produced in this study are available in the following databases: Proteomics data: PRIDE (Proteomics Identifications Database), PXD063572 and PXD063593;

The source data of this paper are collected in the following database record: biostudies:S-SCDT-10_1038-S44321-025-00309-0.

## Peer review information

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

## Acknowledgements

We thank all members of the Proteomics and Signal Transduction Group for help and discussions, and in particular Andre C. Michaelis, Igor Paron, Tim Heymann and Katharina Zettl for technical assistance. We thank Florian Arendt, and Britta Pauli for their clinical assistance and all blood donors who made this research possible. This project was partially supported by the Max Planck Society for the Advancement of Science, the Gates Foundation, the Federal Ministry of Research, Technology and Space Project MSCoreSys (03LW0245) and the German Center for Child and Adolescent Health (DZKJ) under the funding registry 01GL2406D.

## Author contributions

**Kathrin Korff**: Conceptualization; Data curation; Formal analysis; Validation; Investigation; Visualization; Methodology; Writing—original draft; Writing—review and editing. **Johannes B Müller-Reif**: Data curation; Formal analysis; Visualization; Writing—review and editing. **Dorothea Fichtl**: Conceptualization; Resources; Data curation. **Vincent Albrecht**: Data curation; Formal analysis; Visualization; Writing—review and editing. **Alicia-Sophie Schebesta**: Data curation; Formal analysis; Writing—review and editing. **Ericka C M Itang**: Formal analysis; Visualization. **Sebastian Virreira Winter**: Conceptualization; Methodology. **Lesca M Holdt**: Conceptualization; Resources; Methodology. **Daniel Teupser**: Conceptualization; Resources; Validation; Methodology. **Matthias Mann**: Conceptualization; Resources; Supervision; Funding acquisition; Writing—original draft; Project administration; Writing—review and editing. **Philipp E Geyer**: Conceptualization; Resources; Data curation; Software; Formal analysis; Supervision; Validation; Investigation; Visualization; Methodology; Writing—original draft; Writing—review and editing.

Source data underlying figure panels in this paper may have individual authorship assigned. Where available, figure panel/source data authorship is listed in the following database record: biostudies:S-SCDT-10_1038-S44321-025-00309-0.

## Funding

## Disclosure and competing interests statement

Matthias Mann is an indirect investor in Evosep, and Philipp Geyer is a founder and employee of ions.bio. The remaining authors declare no competing interests.

# Expanded View Figures

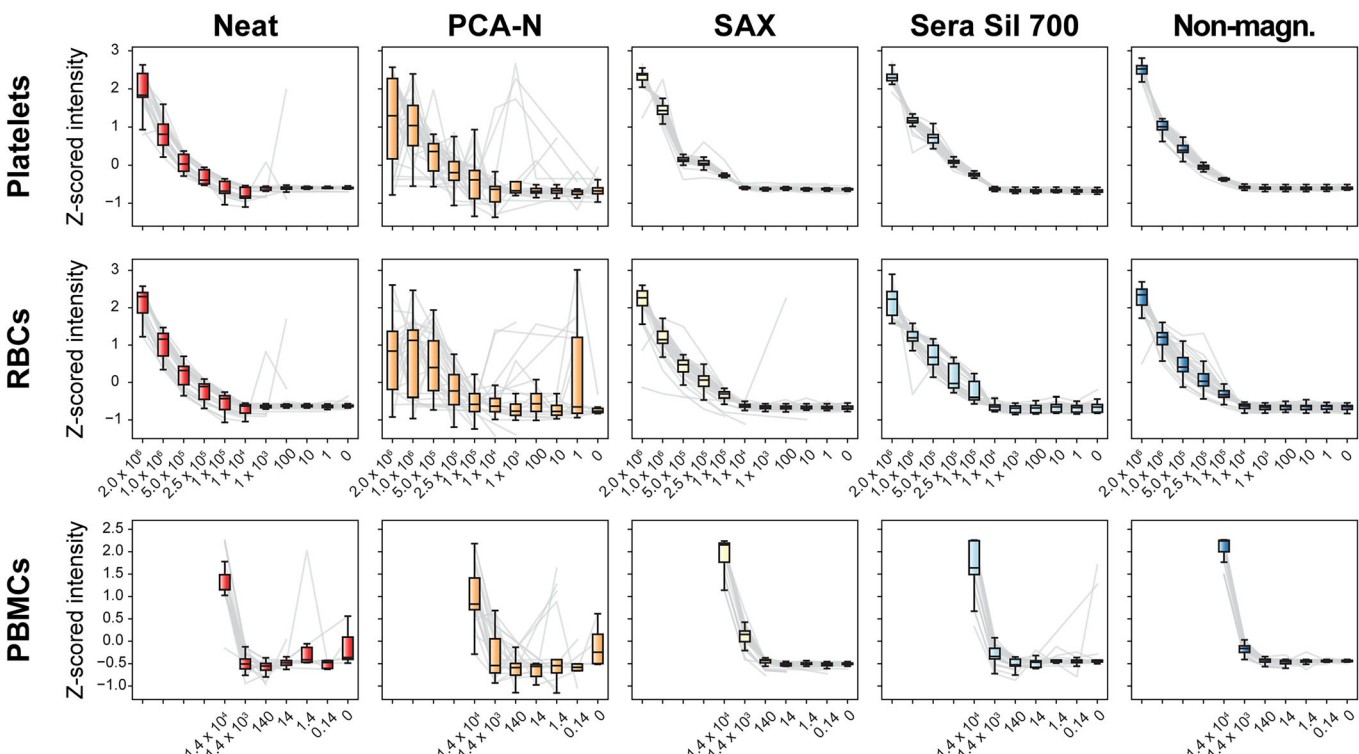

**Figure EV1.  Z-scored intensity profiles of cell-specific quality markers across contamination series in five plasma proteomics workflows.**

Boxplots show Z-scored intensities of the top 30 quality markers for platelets (top row), erythrocytes (middle row), and PBMCs (bottom row) across all contamination levels and workflows. Gray lines represent individual marker proteins, while boxplots display the combined distribution of all 30 markers at each cell concentration. Data were Z-scored per workflow and per protein to allow direct comparison of concentration-dependent patterns.

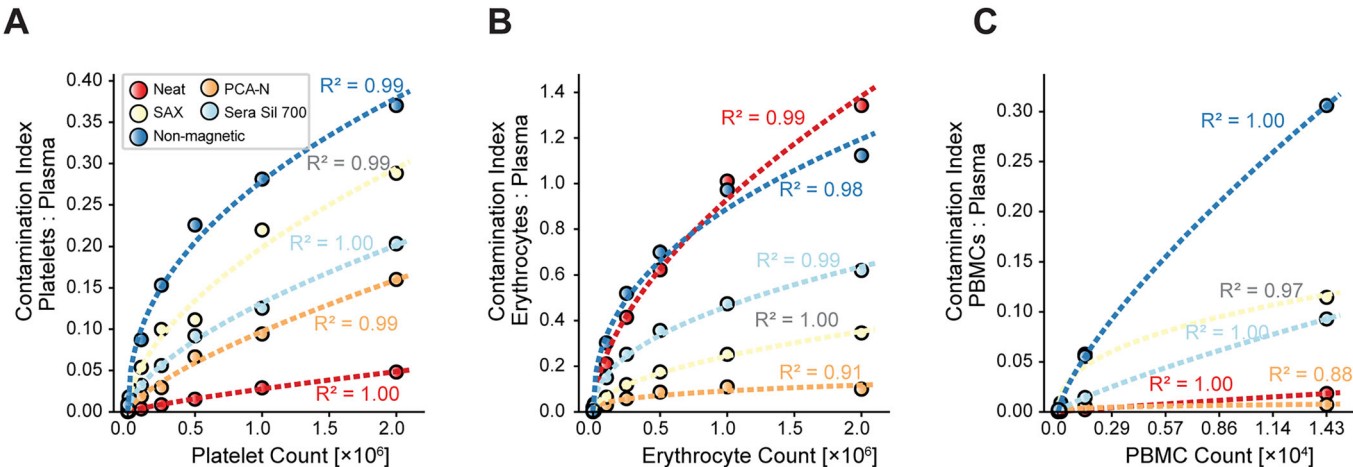

**Figure EV2. Power law model fitting of contamination index data across cell types.**

(A–C) Contamination index versus cell count with power law model fitting for (A) platelets, (B) erythrocytes, and (C) PBMCs across all five workflows. Points represent measured contamination indices at different cell counts, while curved lines show power law model fits. Coefficient of determination ($R^2$) values are displayed for each workflow and cell type.

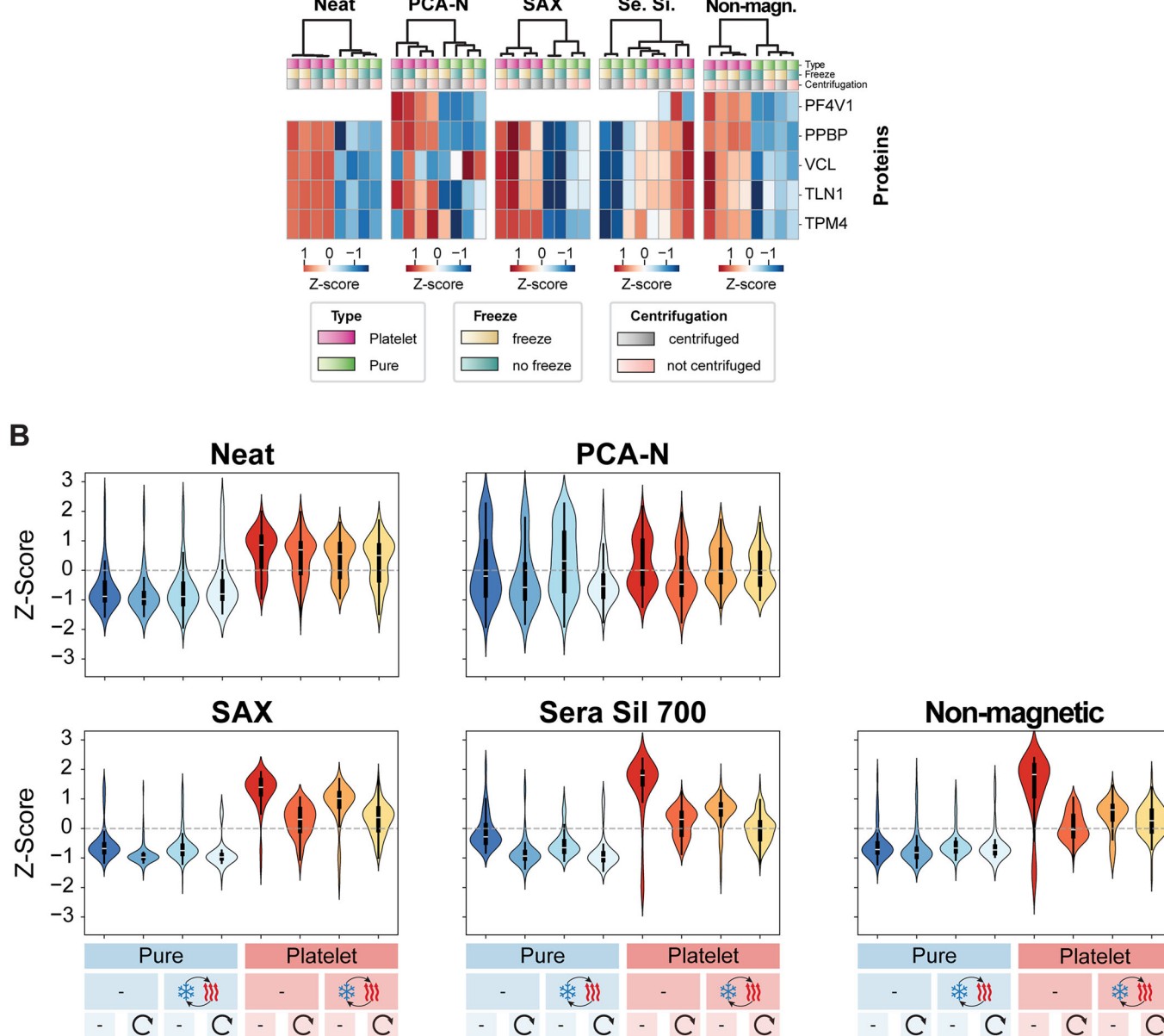

**Figure EV3. Platelet marker analysis across processing conditions.**

(A) Hierarchical clustering of five platelet marker proteins showing workflow-specific responses to processing steps. Data represent four replicates per condition. (B) Violin plots of Z-scored intensities for the top 100 platelet markers across different workflows and processing conditions. Blue: pure plasma; Red: platelet-contaminated. Icons indicate sample type, freeze-thaw status, and centrifugation status.

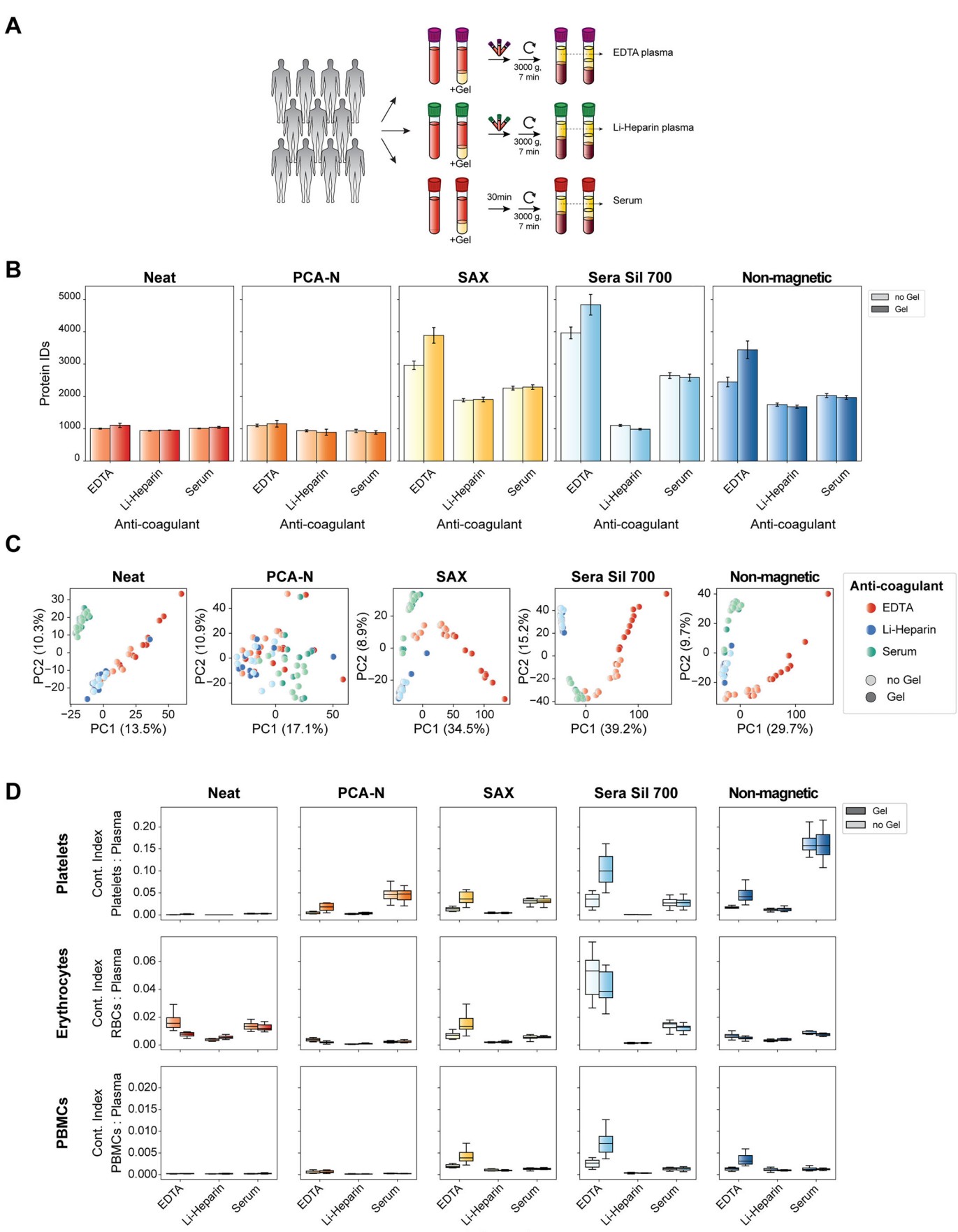

**Figure EV4. Impact of anticoagulants on plasma proteome analysis.**

(A) Experimental design: Blood was collected from 11 healthy individuals into EDTA, Li-Heparin, and serum tubes, with and without gel separators. All samples were centrifuged at 3000×g for 7 min and processed through five proteomic workflows. (B) Number of protein identifications across five workflows for each anticoagulant with and without gel separator tubes. (C) Principal component analysis of proteomics data for each workflow, colored by anti-coagulant type and tube presence. (D) Contamination indices for platelets, erythrocytes, and PBMCs across all workflows and anti-coagulant conditions.

