## [Peer Review File · EMBO Molecular Medicine]

Pre-Analytical Drivers of Bias in Bead-Enriched Plasma Proteomics

Matthias Mann, Kathrin Korff, Johannes Müller-Reif, Dorothea Fichtl, Vincent Albrecht, Alicia-Sophie Schebesta, Ericka Itang, Sebastian Virreira Winter, Lesca Holdt, Daniel Teupser, and Philipp Geyer

Corresponding authors: Matthias Mann (mmann@biochem.mpg.de) , Philipp Geyer (geyer@ions.bio)

Review Timeline:

Submission Date:	8th May 25
Editorial Decision:	5th Jun 25
Revision Received:	4th Jul 25
Editorial Decision:	14th Aug 25
Revision Received:	27th Aug 25
Accepted:	28th Aug 25

Editor: Zeljko Durdevic

Transaction Report:

5th Jun 2025

Dear Dr. Mann,

Thank you for the submission of your manuscript to EMBO Molecular Medicine. We have now received feedback from the three reviewers who agreed to evaluate your manuscript. As you will see from the reports, all three referees are overall supportive of the study raising several important critique points that should be addressed in a major revision. If you would like to discuss further the points raised by the referees, I am available to do so via email or video. Let me know if you are interested in this option.

We would welcome the submission of a revised version within three months for further consideration. Please let us know if you require longer to complete the revision.

I look forward to receiving your revised manuscript.

Yours sincerely,

Zeljko Durdevic

Zeljko Durdevic
Senior Editor
EMBO Molecular Medicine

We require:

- 1) A .docx formatted version of the manuscript text (including legends for main figures, EV figures and tables). Please make sure that the changes are highlighted to be clearly visible.
- 2) Individual production quality figure files as .eps, .tif, .jpg (one file per figure). For guidance, download the 'Figure Guide PDF': (<https://www.embopress.org/page/journal/17574684/authorguide#figureformat>).
- 3) A .docx formatted letter INCLUDING the reviewers' reports and your detailed point-by-point responses to their comments. As part of the EMBO Press transparent editorial process, the point-by-point response is part of the Review Process File (RPF), which will be published alongside your paper.
- 4) A complete author checklist, which you can download from our author guidelines (<https://www.embopress.org/page/journal/17574684/authorguide#submissionofrevisions>). Please insert information in the checklist that is also reflected in the manuscript. The completed author checklist will also be part of the RPF.
- 5) Please note that all corresponding authors are required to supply an ORCID ID for their name upon submission of a revised manuscript.

6) It is mandatory to include a 'Data Availability' section after the Materials and Methods. Before submitting your revision, primary datasets produced in this study need to be deposited in an appropriate public database, and the accession numbers and database listed under 'Data Availability'. Please remember to provide a reviewer password if the datasets are not yet public (see <https://www.embopress.org/page/journal/17574684/authorguide#dataavailability>).

12) Author contributions: You will be asked to provide CRediT (Contributor Role Taxonomy) terms in the submission system. These replace a narrative author contribution section in the manuscript.

13) A Conflict of Interest statement should be provided in the main text.

14) Every published paper now includes a 'Synopsis' to further enhance discoverability. Synopses are displayed on the journal webpage and are freely accessible to all readers. They include a short stand first (maximum of 300 characters, including space) as well as 2-5 one-sentences bullet points that summarizes the paper. Please write the bullet points to summarize the key NEW findings. They should be designed to be complementary to the abstract - i.e. not repeat the same text. We encourage inclusion

of key acronyms and quantitative information (maximum of 30 words / bullet point). Please use the passive voice. Please attach these in a separate file or send them by email, we will incorporate them accordingly.

15) Include a Reagents and Tools Table as part of the Methods section, which can be downloaded from our author guidelines (<https://www.embopress.org/page/journal/17574684/authorguide#structuredmethods>)

**** Reviewer's comments ****

Referee #1 (Comments on Novelty/Model System for Author):

The issue of protocols for plasma proteomics is of interest for specialists in the field and those contemplating doing plasma biomarker discovery work. The approach as detailed is unlikely to be broadly implemented. However the concerns raised regarding protocols for plasma proteomics have merit.

Referee #1 (Remarks for Author):

This paper evaluates issues that may be associated with bead enrichment of plasma proteins for in-depth analysis of the plasma proteome. The authors systematically evaluated five plasma proteomics workflows including three bead-based methods, neat and precipitation protocols. They used spike-ins of low-abundance proteins and critically examined the identification of cellular components as contaminants.

They present findings that bead-based approaches enhance detection of low-abundance proteins but can be highly susceptible to systematic bias from platelet and other cellular contaminants. As a result the findings of thousands of proteins in plasma is in reality not representative of circulating plasma proteins. They provide suggestions for how to mitigate the contribution of cellular contaminants.

The authors are to be commended for raising the concern about contamination of plasma proteins with cellular proteins. However this issue is not strictly limited to bead based assays. The concept of bead based assays to enrich for low abundance proteins has been around for decades. A major issue for this reviewer has been the quantitative accuracy in measuring low abundance proteins. Another issue is whether proteins with complex post-translational modifications may be differentially bound to bead thus limited the potential of bead based assays for discovery and quantitative analysis of the multitude of protein forms. A minor point is around centrifugation. Biobanking sample preps need not include 4 centrifugation steps, as proposed in the manuscript. This seems impractical. Two centrifugations steps would be adequate enough.

These issues need to be addressed by the authors. Overall, the paper would be of interest to a general readership interested in proteomics and biomarkers.

Referee #2 (Comments on Novelty/Model System for Author):

The manuscript demonstrates high technical quality through its comprehensive and systematic evaluation of plasma proteomics workflows. The authors meticulously assess key performance metrics, including proteome coverage, sensitivity to low-abundance protein, and susceptibility to cellular contamination from platelets, erythrocytes and PBMCs. Their investigation into pre-analytical variables, such as sample collection and handling conditions, underscores the critical role of standardized protocols in ensuring reproducibility and reliability in downstream analyses. By addressing these factors, the study highlights how variations in sample processing and handling can potentially lead to data misinterpretation, emphasizing the need for standardized procedures in clinical proteomic workflows.

Regarding the novelty and medical impact, the study offers a thorough comparison of existing plasma proteomics workflows, highlighting their respective advantages and limitations. While it does not introduce a new method or workflow, its systematic approach provides valuable insights for researcher selecting appropriate methodologies for clinical proteomics studies. The medical impact is significant for studies relying on proteomic analyses, but its broader applicability to general medical audiences may be limited to its specialized focus.

The model system employed is appropriate for the study's objectives.

Whether this manuscript with its largely technical focus is appropriate for EMBO Molecular Medicine or a journal more geared towards proteomics is an editorial decision. However, in my opinion the study is rigorous and only minor revisions addressing

the comments to the authors are needed.

Referee #2 (Remarks for Author):

Summary and Overall Assessment:

The manuscript 'Pre-analytical Drivers of Bias in Bead-Enriched Plasma Proteomics' by Korff et al. provides a thorough, comprehensive, and systematic comparison of plasma proteomic workflows. The study specifically evaluates bead-based enrichment methods—designed to enhance plasma proteome depth—against neat plasma and perchloric acid precipitation with neutralization (PCA-N), which aims to deplete abundant plasma proteins.

Key performance metrics assessed include protein yield, proteome coverage, sensitivity to low-abundance proteins, susceptibility to cellular contamination from platelets, erythrocytes and PBMCs. The bead-based approaches achieved the deepest proteome coverage and superior detection of low-abundance proteins, but were more susceptible to cellular contaminants, potentially leading to data misinterpretation.

Importantly, the study investigates the impact of pre-analytical variables such as sample collection and handling conditions. It emphasizes the critical role of standardized plasma collection and processing protocols to ensure reproducibility and reliability in downstream analyses, supporting broader efforts toward harmonization in clinical proteomic workflows.

This methodologically robust study provides insights for researchers in clinical proteomics and related medical fields. The findings have significant implications for biomarker discovery, as variations in sample quality could obscure biological signals, particular in longitudinal, cross-sectional, or multicentric studies.

Comments and Questions for the Authors:

1) Batch Effects and Internal Standards:

How do the authors address batch effects in large-scale sample processing involving multiple biological replicates and conditions? Specifically, do they recommend incorporating internal standards—such as stable isotope-labeled peptides or exogenous protein controls—to monitor digestion efficiency and mass spectrometry variability?

2) Data Analysis and Interpretation:

Can the authors comment on the potential for data misinterpretation introduced by data analysis workflows? Would the field benefit from standardized approaches to data processing and interpretation? Or is such standardization infeasible given the variation in sample types, cohort sizes, and analytical objectives across studies?

3) Workflow Recommendations:

Based on their findings, which workflow(s) do the authors recommend for early-phase discovery (i.e. clinical biomarker) versus large-scale validation studies? In the absence of information about sample collection—particularly for longitudinal and multicentric studies—what recommendations do the authors have for choosing a workflow? For example, beyond centrifugation at 3000 g for 30 minutes, what consideration should be made for biomarker discovery when identification of low-abundance proteins is of high importance? How can biases from the type of anticoagulant used be mitigated if this information is not provided a priori? If provided, is it better to analyze samples in different groups? Is a survey analysis for cell contaminant markers needed to decide the best workflow approach?

To broaden the manuscript's appeal, the authors might consider highlighting specific low-abundance proteins identified through the bead-based enrichment workflow.

4) Summary Table of Conditions

While the study covers an array of conditions, it could benefit of a summary table showing susceptibility to cell contaminants, sample preparation variables, sample collection methods, and proteome depth across all conditions. This would greatly enhance interpretability.

Minor points and Technical Clarifications:

- Figure 1B: please add the names or identities of the cell contaminants shown
- Page 5, third paragraph: it should read 'Figures 2C-E' instead of 'Figure 2C'
- Page 5, last paragraph: it should read 'Figure 2E' instead of 'Figure 2D'
- Figures 2 C-E: standard error bars are stated in the legend but appear to be missing
- Page 7, second paragraph and Figure Labels: it should reference 'Figure 2F-H' instead of 'Figure 2D-F'
- Page 8 and Page 27 (Supplementary Figure 5): why is the number of commonly identified proteins the highest at the 1:100 dilution? Is this result expected, and what might be the underlying explanation?
- Page 8: the yeast sample preparation is referenced to be present in the method section, but it appears to be missing; please include it
- Page 9 Figure 3B and Supplementary figure 5: the colors of SeraSil700 and SAX should be kept consistent across the paper to avoid confusion
- Supplementary Figure 9A: the authors should choose more contrasting colors among conditions to enhance clarity.

Referee #3 (Remarks for Author):

The study is of critical importance, particularly at a time when the number of protein identifications is still often mistaken for the quality of a method or analysis. It provides a thorough investigation into the concerning impact of residual blood cells in plasma across several widely used workflows and offers insight into why bead-based enrichment results vary so significantly between laboratories. This work will raise awareness in the community and may encourage commercial vendors to evaluate their products more rigorously for robustness against sample quality issues and cellular contamination. If this problem is not recognized and addressed, studies risk being fundamentally flawed.

Recommendations/Comments:

"Rescue" / "Partial rescue"

While centrifugation may help harmonize samples of differing quality or contamination levels, this has not been demonstrated. Platelet-enriched samples, even after centrifugation, still show elevated protein identifications and diverge from neat plasma in PCA analysis (Figure 5). Additionally, it remains unclear how proteins from ruptured cells-potentially occurring during collection or preparation-interact with bead surfaces. I recommend softening or rephrasing this claim.

"PCA-N provides unique resistance to erythrocyte-derived proteins"

The authors' own recently introduced method is positioned as the preferred approach. While Figure 1 suggests robustness to erythrocyte contamination, Figures S2 and S4 show that highly abundant erythrocyte proteins still influence the results. Moreover, PCA-N doesn't consistently yield substantially higher protein identifications, exhibits greater variance, and appears more susceptible to platelet contamination-which is harder to remove than erythrocytes. Therefore, I suggest avoiding positioning any single method as superior.

Page 3: "Exploring the biological mechanism". I was expecting a biological interpretation of why certain proteins are enriched or depleted under different conditions. The phrase "biological mechanism" suggested a functional or mechanistic explanation that I feel, I did not get.

Given the workflow's sensitivity to small cellular contaminants, a statement clarifying how residual intact yeast cells were excluded would be useful.

Page 11: "Contribution of freeze-thawing was minimal"

While freeze-thawing may not alleviate the effect of cellular contamination it still had a strong effect on the variance in the data, especially when samples were not centrifuged. I would therefore recommend to rephrase this statement.

A recommendation on dealing with acquired or already published datasets would be useful. The authors present contamination indices but it is unclear how they can/should be used: excluding samples, forming experimental sub-groups, is correction possible?

Wishlist: Enrichment/Depletion analysis of the samples after the 30-minute centrifugation - similar or in combination to Figure 1E. This could offer a cleaner view of which proteins are truly plasma-derived.

It seems the "pure" plasma used in the initial experiments may still contain residual cells. For example, in Figure 5B (pure plasma barplot) and Figure 5C (Sera Sil 700), there are hints of platelet contamination, as post-centrifugation samples appear to shift toward the contaminated profile.

Point-by-point response to reviewer comments for “Pre-Analytical Drivers of Bias in Bead-Enriched Plasma Proteomics”

We thank the reviewers for their thoughtful and constructive feedback, which significantly improved the manuscript. All three recognized the importance of our study, particularly the need for standardized plasma proteomics workflows and increased awareness of contamination-related artifacts.

Reviewer 1 supported the study's focus on contamination but raised valid concerns about the quantitative accuracy of bead-based methods for low-abundance proteins, potential biases from post-translational modifications, and the need to avoid overgeneralizing contamination as a bead-specific issue.

Reviewer 2 described the study as rigorous and technically strong, suggesting only minor revisions such as correcting figure references, addressing missing error bars and yeast methods, and providing workflow recommendations and a summary comparison table.

Reviewer 3 was highly supportive of the study's relevance but emphasized the need to clarify the limits of "rescue" by centrifugation, temper claims about PCA-N, and offer practical guidance on using contamination indices in retrospective datasets. They also requested clarification on the removal of intact yeast cells.

We have addressed each of these points in detail and revised the manuscript accordingly.

In the following point-by-point response, we outline our revisions and the rationale behind them. We have used the following color scheme to distinguish elements:

- Black: Reviewer comments
- Blue: Our explanation and response
- Green: Text added or modified in the revised manuscript

We hope the revised manuscript now meets the requirements for publication in EMBO Molecular Medicine, as we have significantly improved the manuscript in response to the reviewers' input. Thank you for your consideration.

In response to the reviewers' feedback and to strengthen our manuscript, we have conducted several additional analyses:

- Quantitative precision profiling across ten protein abundance categories, confirming decreased reproducibility for low-abundance proteins across all workflows (**Response Figures 1–3; Appendix Figure S1E**).
- A literature survey of centrifugation protocols in ~300 plasma proteomics studies, revealing substantial heterogeneity and underreporting of critical parameters (**Response Figure 4**).
- Overlap analysis of low-abundance proteins between workflows, demonstrating that bead methods access a distinct subset poorly captured by neat plasma (**Response Figure 5; Appendix Figure S1A**).
- Functional enrichment analysis of ~850 proteins uniquely identified by bead-based methods, revealing overrepresentation of extracellular, tissue-derived proteins (**Response Figure 6; Appendix Figure S1B-D**).
- A new physicochemical analysis comparing enriched vs. depleted proteins within each workflow, showing that bead-based methods preferentially enrich smaller, shorter, and more positively charged proteins (**new Figure 3E**).
- Enrichment/depletion analysis before and after centrifugation, showing that contamination removal not only reduces carryover but also enhances detection of low-abundance proteins in bead workflows (**Response Figure 8; Appendix Figure S9**).
- Workflow-specific contamination marker analyses, revealing distinct marker compositions across workflows while confirming the robustness of our universal contamination indices derived from neat plasma (**Appendix Figure S4**).

Reviewer #1

Comments on Novelty/Model System for Author:

The issue of protocols for plasma proteomics is of interest for specialists in the field and those contemplating doing plasma biomarker discovery work. The approach as detailed is unlikely to be broadly implemented. However the concerns raised regarding protocols for plasma proteomics have merit.

We thank the reviewer for this thoughtful assessment. We agree that broad adoption of our protocols is unlikely. Our primary aim is to raise awareness within the plasma proteomics community of the substantial impact of pre-analytical factors, such as centrifugation, anticoagulant choice, and handling, on proteome composition. While we recognize that changing established protocols may not be practical in all settings, we believe it is essential to interpret high protein identification numbers (e.g., 6,000+) in the context of potential contamination, particularly when using bead-based enrichment. Our goal is to encourage more informed evaluation of plasma proteomics data and to promote quality control practices, even if implementation varies across laboratories.

Remarks for Author:

This paper evaluates issues that may be associated with bead enrichment of plasma proteins for in-depth analysis of the plasma proteome. The authors systematically evaluated five plasma proteomics workflows including three bead-based methods, neat and precipitation protocols. They used spike-ins of low-abundance proteins and critically examined the identification of cellular components as contaminants.

They present findings that bead-based approaches enhance detection of low-abundance proteins but can be highly susceptible to systematic bias from platelet and other cellular contaminants. As a result the findings of thousands of proteins in plasma is in reality not representative of circulating plasma proteins. They provide suggestions for how to mitigate the contribution of cellular contaminants.

The authors are to be commended for raising the concern about contamination of plasma proteins with cellular proteins. However this issue is not strictly limited to bead based assays.

We thank the reviewer for this important point and fully agree that contamination is not exclusive to bead-based workflows. Our data show that the neat and PCA-N workflows are also susceptible to platelet and PBMC contamination, although to a lesser degree.

Nonetheless, our systematic comparison demonstrates that bead-based workflows are substantially more sensitive to cellular contamination, often yielding 6,000–8,000 protein identifications under contaminated conditions, far exceeding the numbers observed with neat or PCA-N workflows. This differential susceptibility is a key finding and has important implications for workflow selection based on sample quality.

In response, we have added a statement to the Discussion emphasizing that all workflows are affected by contamination and that quality monitoring is essential regardless of the method used:

“The neat workflow occupies a middle ground with moderate vulnerability but fewer identifications. The latter two technologies offer advantages of automation, low additional

costs, and ease of use. Importantly, while these workflows differ in their degree of susceptibility, all plasma proteomics approaches are affected by cellular contamination to some extent, making quality monitoring and assessment of pre-analytical biases essential regardless of the chosen methodology.”

The concept of bead based assays to enrich for low abundance proteins has been around for decades. A major issue for this reviewer has been the quantitative accuracy in measuring low abundance proteins.

The reviewer raises an important question about the quantitative accuracy of low-abundance protein measurements. To address this concern, we performed additional analyses across all five workflows.

We first evaluated quantitative precision across abundance ranges by measuring normal plasma samples in quadruplicates and calculating coefficients of variation (CVs) for proteins grouped into ten abundance categories (**Response Figure 1**). As expected, CVs increased for lower-abundance proteins in all workflows, including neat plasma (**Appendix Figure S1E**), demonstrating that reduced precision at low abundance is not specific to bead-based enrichment but is a general limitation of mass spectrometry-based proteomics.

To directly compare across workflows, we selected identical high- and low-abundance proteins based on neat plasma ranking (**Response Figure 2, Appendix Figure S1E**) CVs for low-abundance proteins were comparable across workflows, with only marginally higher variability in bead-based methods. High-abundance proteins showed consistent CVs across most workflows, though PCA-N displayed slightly elevated CVs in that range, possibly due to its depletion strategy and increased variance overall. We now clarify this pattern in the revised text.

We further validated these findings with yeast spike-in experiments (**Response Figure 3**). By analyzing the top 100 yeast proteins across a dilution series (from 50% down to 0.0001% yeast in plasma), we observed that CVs systematically increased as spike-in concentration decreased. This pattern was consistent across neat, PCA-N, and bead-based workflows and reflects the expected decrease in reproducibility at lower protein abundances due to reduced signal-to-noise ratios.

Regarding relative abundance changes: Bead-based enrichment (and PCA-N) alters the rank order of proteins due to differential enrichment and depletion effects (e.g., **Figure 1E**). However, these effects are consistent within workflows, so fold-change relationships across experimental conditions are preserved. Thus, while absolute abundances may shift, relative quantification remains valid.

We agree this is an important consideration that affects all plasma proteomics workflows, beyond the issue of contamination.

“Quantitative precision analysis across abundance ranges confirms that reduced precision for low-abundance proteins is a general feature of MS-based proteomics, affecting all workflows degree (**Appendix Figure S1E**).”

Response Figure 1 - Quantitative precision across protein abundance categories

CV analysis across five plasma proteomics workflows. Left panels show rank abundance plots for each workflow with proteins colored by abundance category (1-10, from high to low abundance). Right panels display CV distributions for each abundance category. Data derived from quadruplicate measurements of normal plasma samples.

Response Figure 2 - **Direct comparison of high vs. low-abundance protein quantitative precision.**

Coefficient of variation comparison for identical protein sets across workflows. Analysis focuses on high-abundance proteins (Category 1) and low-abundance proteins (Category 10) as defined by neat plasma ranking. Violin plots show CV distributions

Response Figure 3 - **Yeast spike-in validation of abundance-dependent quantitative precision**

CV analysis of the top 100 most abundant yeast proteins across dilution series (1:1 to 1:100000 yeast:plasma ratios) for each workflow. The dilution series simulates the transition from high-abundance to low-abundance proteins in a controlled manner.

Another issue is whether proteins with complex post-translational modifications may be differentially bound to bead thus limited the potential of bead based assays for discovery and quantitative analysis of the multitude of protein forms.

We thank the reviewer for highlighting this important point regarding the potential for proteins with complex post-translational modifications (PTMs) to bind differentially to bead surfaces. We have added a statement in the Discussion saying that such modifications may influence binding efficiency, potentially affecting both the discovery and quantification of diverse protein forms. This

introduces another layer of selectivity bias beyond cellular contamination and emphasizes the need for further research into corona formation and bead-surface interactions.

“Thus, the increased sensitivity is real but comes at the cost of greater vulnerability to bias when working with non-ideal specimens. An additional limitation of bead-based enrichment is that proteins with complex post-translational modifications may bind differentially to bead surfaces, potentially affecting both the discovery and quantitative analysis of diverse protein forms and introducing an additional layer of selectivity bias beyond cellular contamination effects.”

A minor point is around centrifugation. Biobanking sample preps need not include 4 centrifugation steps, as proposed in the manuscript. This seems impractical. Two centrifugations steps would be adequate enough.

We thank the reviewer for this comment and the opportunity to clarify our approach. We did not use four sequential centrifugation steps. Rather, we compared four different single-spin conditions (500g for 7 min, 1,000g for 7 min, 3,000g for 7 min, and 3,000g for 30 min) by splitting blood samples and applying one condition per aliquot to determine the optimal single-step protocol. We understand this may have been misinterpreted and have revised the text accordingly.

“We collected blood from 11 healthy individuals into standard EDTA tubes and EDTA gel separator tubes. Each tube was then centrifuged once using one of four different conditions: 500g for 7 min, 1,000g for 7 min, 3,000g for 7 min, and 3,000g for 30 min. (Figure 6A)”

To assess real-world feasibility, we also reviewed ~300 plasma proteomics publications (**Response Figure 4**). Among studies reporting centrifugation parameters, conditions varied widely: centrifugal force ranged from 1,000–3,000 ×g, durations were typically 10–15 minutes, and only 8 studies reported double centrifugation (needed for platelet-poor plasma). Notably, 43% did not document temperature, and 39% processed at 4°C, despite the risk of platelet activation.

We agree that implementing a 30-minute spin may not be feasible for all labs or biobanks. Nonetheless, based on our findings, we recommend a single centrifugation at 3,000g for 7–30 minutes to achieve the best sample quality.

Importantly, even archived samples processed under suboptimal conditions may benefit from an additional centrifugation prior to preparation. While this does not fully eliminate contamination, our ‘rescue’ experiments show that it can significantly reduce platelet and other cellular carryover, improving data quality in downstream proteomics.

Response Figure 4 - **Heterogeneity in preanalytical plasma processing across plasma proteomics studies.**

Data are based on a literature review of approximately 300 plasma proteomics studies.

(A) Anticoagulant types reported in 106 studies, including EDTA, heparin and citrate formulations.

(B) Centrifugation temperatures during single-spin plasma preparation as reported in 80 studies.

(C, D) Distribution of centrifugation times and centrifugal forces for single-spin protocols (n = 80).

(E, F) Centrifugation times and speeds for first and second spins in studies reporting double-spin procedures (n = 8).

These issues need to be addressed by the authors. Overall, the paper would be of interest to a general readership interested in proteomics and biomarkers.

We thank the reviewer for their constructive feedback and positive assessment of our study's relevance to the proteomics and biomarker community. By addressing all points raised we believe

the revisions have strengthened the manuscript's clarity and practical value. We appreciate the recognition that this work will be of broad interest to researchers in plasma proteomics and biomarker discovery.

Reviewer #2

Comments on Novelty/Model System for Author:

The manuscript demonstrates high technical quality through its comprehensive and systematic evaluation of plasma proteomics workflows. The authors meticulously assess key performance metrics, including proteome coverage, sensitivity to low-abundance protein, and susceptibility to cellular contamination from platelets, erythrocytes and PBMCs. Their investigation into pre-analytical variables, such as sample collection and handling conditions, underscores the critical role of standardized protocols in ensuring reproducibility and reliability in downstream analyses. By addressing these factors, the study highlights how variations in sample processing and handling can potentially lead to data misinterpretation, emphasizing the need for standardized procedures in clinical proteomic workflows.

We sincerely thank the reviewer for their positive assessment, for recognizing the systematic and comprehensive nature of our study and the acknowledgement of our main objective.

To expand on the reviewer's point about data misinterpretation: we emphasize that pre-analytical variation, particularly cellular contamination, can lead to incorrect data generation at the outset. In such cases, protein identifications may not reflect the circulating plasma proteome, compromising datasets before downstream analysis even begins.

We agree that standardized protocols are essential for reproducibility in clinical proteomics. While full standardization is not always feasible across routine settings, our goal is to raise awareness of these critical pre-analytical factors and to provide guidance that enables informed choices about workflow selection and data interpretation across diverse laboratory contexts.

Regarding the novelty and medical impact, the study offers a thorough comparison of existing plasma proteomics workflows, highlighting their respective advantages and limitations. While it does not introduce a new method or workflow, its systematic approach provides valuable insights for researcher selecting appropriate methodologies for clinical proteomics studies. The medical impact is significant for studies relying on proteomic analyses, but its broader applicability to general medical audiences may be limited to its specialized focus.

The model system employed is appropriate for the study's objectives.

Whether this manuscript with its largely technical focus is appropriate for EMBO Molecular Medicine or a journal more geared towards proteomics is an editorial decision. However, in my opinion the study is rigorous and only minor revisions addressing the comments to the authors are needed.

While we do not introduce a new method, the reviewer agrees that our systematic comparison offers valuable insights for researchers selecting appropriate workflows in clinical proteomics. Although the workflows examined originated within the proteomics community, they have already been widely adopted for biological discovery and biomarker research. We believe that raising awareness of pre-analytical challenges will benefit a broad range of researchers. By systematically evaluating contamination susceptibility under real-world conditions, this study addresses a key gap in understanding workflow performance and contributes to more reliable biomarker discovery.

Regarding journal scope, we consider EMBO Molecular Medicine highly appropriate, especially since this work extends our previous publication in the journal (Geyer et al., 2019, EMBO Mol Med 11: e10427), which established plasma quality marker panels. Our current study builds on that foundation by evaluating how various workflows handle sample quality, making it a natural continuation of that research within the journal's mission.

Remarks for Author:

Summary and Overall Assessment:

The manuscript 'Pre-analytical Drivers of Bias in Bead-Enriched Plasma Proteomics' by Korff et al. provides a thorough, comprehensive, and systematic comparison of plasma proteomic workflows. The study specifically evaluates bead-based enrichment methods-designed to enhance plasma proteome depth-against neat plasma and perchloric acid precipitation with neutralization (PCA-N), which aims to deplete abundant plasma proteins.

Key performance metrics assessed include protein yield, proteome coverage, sensitivity to low-abundance proteins, susceptibility to cellular contamination from platelets, erythrocytes and PBMCs. The bead-based approaches achieved the deepest proteome coverage and superior detection of low-abundance proteins, but were more susceptible to cellular contaminants, potentially leading to data misinterpretation.

Importantly, the study investigates the impact of pre-analytical variables such as sample collection and handling conditions. It emphasizes the critical role of standardized plasma collection and processing protocols to ensure reproducibility and reliability in downstream analyses, supporting broader efforts toward harmonization in clinical proteomic workflows.

This methodologically robust study provides insights for researchers in clinical proteomics and related medical fields. The findings have significant implications for biomarker discovery, as variations in sample quality could obscure biological signals, particular in longitudinal, cross-sectional, or multicentric studies.

We thank the reviewer for clearly capturing the systematic nature of our study and the key finding, namely, the current trade-off between proteome depth and susceptibility to cellular contamination across workflows. We also appreciate the recognition of its broader relevance for biomarker discovery and the need for standardized protocols.

To expand on the point about data misinterpretation, we want to emphasize that cellular contamination affects the data generation itself, leading to protein identifications that may not accurately reflect the circulating plasma proteome. This is particularly critical in longitudinal, cross-sectional, or multicentric studies, where variation in sample quality can obscure true biological signals and compromise downstream analyses.

Comments and Questions for the Authors:

1) Batch Effects and Internal Standards:

How do the authors address batch effects in large-scale sample processing involving multiple biological replicates and conditions? Specifically, do they recommend incorporating internal

standards-such as stable isotope-labeled peptides or exogenous protein controls-to monitor digestion efficiency and mass spectrometry variability?

This is an important question regarding quality control in large-scale studies. While batch effects and internal standards were not the primary focus of our contamination-focused study, we agree that that is essential when differences between preparation plates or instruments are observed.

In our standard workflows, we typically do not use isotope-labeled proteins or peptides. Instead, we ensure consistency by using the same reagent batches, preparing buffers in bulk, standardizing digestion protocols, and monitoring technical metrics such as missed cleavage rates. For quality control, we routinely include at least two QC samples (e.g., pooled plasma) per preparation plate to assess variability in sample preparation and MS performance.

These measures are particularly important when applying the contamination assessment strategies outlined in our study. Our contamination indices and quality marker panels can help distinguish technical variability from biological or contamination-related bias, improving confidence in data interpretation.

2) Data Analysis and Interpretation:

Can the authors comment on the potential for data misinterpretation introduced by data analysis workflows? Would the field benefit from standardized approaches to data processing and interpretation? Or is such standardization infeasible given the variation in sample types, cohort sizes, and analytical objectives across studies?

Regarding data analysis standardization, while our study focused on pre-analytical contamination, we agree that this broader issue is highly relevant to the field. In our view, the risk of misinterpretation from modern data analysis workflows is relatively low. Commonly used tools, such as DIA-NN, Python-based pipelines, and other community-adopted software, embody shared principles that contribute to a degree of informal standardization. Moreover, the proteomics community's open-science practices of data sharing allow for independent reanalysis and verification.

That said, even the most robust analytical pipelines cannot correct for compromised input caused by cellular contamination. This underscores the importance of addressing pre-analytical factors during sample preparation. We therefore recommend that researchers cross-check significant hits against contamination marker panels before drawing biological conclusions.

3) Workflow Recommendations:

Based on their findings, which workflow(s) do the authors recommend for early-phase discovery (i.e. clinical biomarker) versus large-scale validation studies? In the absence of information about sample collection-particularly for longitudinal and multicentric studies-what recommendations do the authors have for choosing a workflow? For example, beyond centrifugation at 3000 g for 30 minutes, what consideration should be made for biomarker discovery when identification of low-abundance proteins is of high importance? How can biases from the type of anticoagulant used be mitigated if this information is not provided a priori? If provided, it is better to analyze samples

in different groups? Is a survey analysis for cell contaminant markers needed to decide the best workflow approach?

We thank the reviewer for these highly relevant questions that directly address the practical implementation of our findings. These are exactly the considerations researchers face when designing proteomics studies.

Workflow recommendations are challenging to provide as they depend heavily on the study's aims. The neat plasma workflow remains the ground truth, as every protein is measured in the exact concentrations at which it occurs in plasma, preserving the precise ratios between different proteins before they undergo MS analysis. However, when greater depth in protein identification is required, especially when low-abundance proteins are of high importance, PCA-N or bead-based workflows become necessary for discovery studies.

We find that rigorous data evaluation is as important as workflow selection. We recommend a three-step approach: First, analyze the quality of each sample using contamination indices, removing any outlier samples with excessive contamination from the dataset. Second, evaluate whether there is systematic bias in quality markers between cases and controls. Third, assess whether significant hits are contamination panel members or show high correlation with contamination markers.

Regarding anticoagulant considerations, our previous work (Geyer et al., 2019, EMBO Mol Med 11: e10427) demonstrated substantial differences between anticoagulants, especially between plasma and serum, which can complicate data analysis. However, quality panels can help determine whether significant hits represent genuine biological effects or anticoagulant-related artifacts. When different anticoagulants were used and this wasn't equally distributed between cases and controls, separate analysis of these groups is advisable.

A survey analysis to decide upon a workflow is not necessarily needed, as sample preparation takes similar time regardless of scale (10 vs 100 samples), making surveys time-consuming without proportional benefit. Additionally, contamination often affects only a subset of samples rather than entire studies, limiting survey utility. The optimal approach, when uncertain about workflow selection, would be to employ all three methods - neat plasma, PCA-N, and bead-based workflows - providing comprehensive coverage and internal validation.

To broaden the manuscript's appeal, the authors might consider highlighting specific low-abundance proteins identified through the bead-based enrichment workflow.

We thank the reviewer for this suggestion. To highlight biologically relevant low-abundance proteins uniquely accessed by bead-based workflows, we performed an overlap analysis between low-abundance proteins (categories 7–10, **Response Figure 1**) detected by each bead method and those found in neat plasma (**Response Figure 5 and Appendix Figure S1A**). This showed that less than 25% of the lowest 30% of the bead-detected low-abundance proteins overlapped with neat plasma. Most of these ranked at the lowest end of the neat plasma abundance distribution, as seen in our rank-abundance plots (highlighted in red). This supports the notion that bead methods access proteins at or near the detection limits of neat plasma workflows.

Response Figure 5 - **Overlap analysis of low-abundance proteins between workflows**

Venn diagrams and rank abundance plots showing overlap between low-abundance proteins (categories 7-10) detected by each workflow and all proteins identified in neat plasma. Right: Neat plasma rank abundance plots with overlapping proteins with the low abundant bead-based / PCA-N proteins highlighted in red.

To further investigate this distinct protein set, we performed enrichment analysis on ~850 proteins uniquely identified by bead methods (**Response Figure 6**). Cellular component analysis showed strong enrichment for extracellular regions, vesicles, and exosomes. Subcellular localization confirmed this pattern, with overrepresentation of secretory granules and extracellular compartments. Tissue expression analysis revealed enrichment for proteins originating from liver, digestive glands, blood, and the hematopoietic system. These results indicate that bead-based workflows preferentially capture tissue-specific leakage proteins and secreted factors circulating at low concentrations, molecules that are often invisible to neat plasma analysis. Among them, we found cytokine-related proteins such as ILF3 and IL1RL1 exclusively detected in the bead-based workflows, further emphasizing their utility in uncovering biologically relevant low-

abundance proteins. This highlights their potential utility in detecting subtle pathological changes and identifying tissue-specific biomarkers.

Response Figure 6 - **Functional enrichment analysis of proteins uniquely detected by bead-based methods**

Enrichment analysis of approximately 850 proteins uniquely detected by bead-based workflows compared to neat plasma.

(A) Cellular component analysis.

(B) Subcellular localization analysis.

(C) Tissue expression analysis.

Dot size represents gene count, color intensity indicates statistical significance (FDR).

“These differential enrichment patterns directly determine which proteins can be reliably detected and quantified in plasma samples and provide guidance for workflow selection based on target protein characteristics and research objectives. We found that a small proportion of the low-abundance bead-detected proteins overlap with neat plasma (< 25% of the lowest 30% of the bead proteome), and these proteins are at the lowest-abundance edge of the neat plasma proteome (**Appendix Figure S1A**). An enrichment analysis of bead specific protein populations revealed tissue-specific markers, extracellular components, and proteins from various subcellular compartments. This indicates that bead-based methods capture biologically relevant secreted and tissue-derived proteins that circulate at low concentrations (**Appendix Figure S1B-D**).”

4) Summary Table of Conditions

While the study covers an array of conditions, it could benefit of a summary table showing susceptibility to cell contaminants, sample preparation variables, sample collection methods, and proteome depth across all conditions. This would greatly enhance interpretability.

We agree and provide such a table in **Response Table 1** and as **EV Table 1**.

	Neat	PCA-N	SAX	Sera Sil 700	Non-magnetic
Proteome depth	Low	Medium	High	High	High
Variability	Low	Medium	Low	Low	Low
Sample preparation duration	Low	Medium	Medium	Medium	High
Platelet susceptibility	Low	Medium	High	High	High
Erythrocyte susceptibility	Medium	Low	High	High	High
PBMC susceptibility	Medium	Medium	High	High	High

Response Table 1 - Summary of workflow characteristics across all tested conditions.

Comparison of the five evaluated plasma proteomics workflows across key performance and susceptibility criteria. Categories include proteome depth, technical variability, sample preparation duration, and susceptibility to cellular contamination (platelets, erythrocytes, and PBMCs). Values are semi-quantitative and based on experimental results. "Low," "Medium," and "High" are relative ratings to support qualitative interpretation and workflow comparison.

Minor points and Technical Clarifications:

- Figure 1B: please add the names or identities of the cell contaminants shown

Done

- Page 5, third paragraph: it should read 'Figures 2C-E' instead of 'Figure 2C'

Done

- Page 5, last paragraph: it should read 'Figure 2E' instead of 'Figure 2D'

Done

- Figures 2 C-E: standard error bars are stated in the legend but appear to be missing

For visual clarity and to avoid overcomplicating these plots, we intentionally did not include error bars in Figures 2C-E, as they would make the figures difficult to interpret given the multiple overlapping data series. We have corrected the figure legend to accurately reflect that error bars are not shown in these panels to maintain figure readability.

- Page 7, second paragraph and Figure Labels: it should reference 'Figure 2F-H' instead of 'Figure 2D-F'

Done

- Page 8 and Page 27 (Supplementary Figure 5): why is the number of commonly identified proteins the highest at the 1:100 dilution? Is this result expected, and what might be the underlying explanation?

We thank the reviewer for this question, but we believe there may be a misunderstanding. As shown in Appendix Figure S5, the number of commonly identified proteins decreases with increasing dilution, as expected. The 1:2 dilution shows the highest number of commonly identified proteins (1,128), which decreases progressively to 30 proteins at the 1:10000 dilution. This pattern is biologically logical - higher yeast concentrations provide more detectable proteins that can be consistently identified across all workflows, while extreme dilutions result in fewer proteins being reliably detected by all methods. To make this more clear, we have rephrased in the revised manuscript:

“As expected, the number of proteins commonly identified across all workflows decreased with increasing dilution—from 1,128 at the 1:2 ratio to only 30 at 1:10,000—reflecting the expected concentration-dependent loss of detectable signal (Appendix Figure S5).”

- Page 8: the yeast sample preparation is referenced to be present in the method section, but it appears to be missing; please include it

Thank you for catching this. We now provide this description:

“Yeast protein preparation

Yeast proteins for spike-in experiments were prepared from *Saccharomyces cerevisiae* strain BY4741. Briefly, 2 mg of yeast was mixed with 200 μ L of PBS and subjected to bead milling to disrupt the cells and extract yeast proteins. To address the potential for residual intact yeast cells, the lysate was subjected to high-speed centrifugation at 16,000 \times g for 20 minutes. This rigorous centrifugation effectively pellets intact cells and large cellular debris, which are excluded from the resulting supernatant that contains the extracted yeast proteins. The supernatant was carefully collected without disturbing the pellet and used for generating the yeast dilution series in plasma samples.”

- Page 9 Figure 3B and Supplementary figure 5: the colors of SeraSil700 and SAX should be kept consistent across the paper to avoid confusion

Done

- Supplementary Figure 9A: the authors should choose more contrasting colors among conditions to enhance clarity.

Done

Reviewer #3

Remarks for Author:

The study is of critical importance, particularly at a time when the number of protein identifications is still often mistaken for the quality of a method or analysis. It provides a thorough investigation into the concerning impact of residual blood cells in plasma across several widely used workflows and offers insight into why bead-based enrichment results vary so significantly between laboratories. This work will raise awareness in the community and may encourage commercial vendors to evaluate their products more rigorously for robustness against sample quality issues and cellular contamination. If this problem is not recognized and addressed, studies risk being fundamentally flawed.

We sincerely thank the reviewer for this strong endorsement, agreeing on the importance of our study. The reviewer articulates our central concern, that high protein identification numbers are too often equated with method quality, overlooking the confounding effect of contamination. We particularly appreciate the insight that our findings help explain variability in bead-based enrichment results across laboratories, likely due to differences in sample quality rather than methodological inconsistency.

Recommendations/Comments:

"Rescue" / "Partial rescue"

While centrifugation may help harmonize samples of differing quality or contamination levels, this has not been demonstrated. Platelet-enriched samples, even after centrifugation, still show elevated protein identifications and diverge from neat plasma in PCA analysis (Figure 5). Additionally, it remains unclear how proteins from ruptured cells—potentially occurring during collection or preparation—interact with bead surfaces. I recommend softening or rephrasing this claim.

We that 'rescue' may be somewhat misleading' As the reviewer correctly points out, our data do not demonstrate full restoration of sample quality. Figure 5 shows that platelet-enriched samples, even after centrifugation, still have elevated protein identifications and remain distinct from neat plasma in PCA analysis. Our results indicate that additional centrifugation improves sample quality by reducing contamination, but do not fully revert the proteome profile to that of clean plasma. We also appreciate the reviewer's point regarding proteins from ruptured cells, which may still interact with bead surfaces and contribute to residual contamination, an issue mechanical separation cannot fully resolve.

In response, we have revised the manuscript to use more accurate terms such as 'contamination mitigation', 'partial improvement', and 'contamination reduction' instead of 'rescue'. We now explicitly state that the proteomes of centrifuged samples remain distinct from clean plasma, even while showing reduced contamination.

“Given that already collected plasma samples can have significant platelet contamination, which is readily apparent in sensitive plasma proteomics workflows, especially bead-based ones, we asked to what extent this could be mitigated.”

“In all cases except PCA-N, the PCA clearly separated centrifuged and non-centrifuged samples in the rescue experiments, supporting the benefit of this step. However, it should be noted that even after centrifugation, contaminated samples maintained distinct proteome profiles and did not fully return to the baseline of pure plasma samples.”

“This reduction suggested improvement in sample quality, although samples remained distinct from pure plasma. This pattern was consistent across all workflows, with bead-based methods showing the most pronounced reduction in contamination, which was also evident at the single quality marker level (**Fig EV3**).”

“PCA-N provides unique resistance to erythrocyte-derived proteins”

The authors' own recently introduced method is positioned as the preferred approach. While Figure 1 suggests robustness to erythrocyte contamination, Figures S2 and S4 show that highly abundant erythrocyte proteins still influence the results. Moreover, PCA-N doesn't consistently yield substantially higher protein identifications, exhibits greater variance, and appears more susceptible to platelet contamination-which is harder to remove than erythrocytes. Therefore, I suggest avoiding positioning any single method as superior.

We agree with this fair critique. It was not our intention to position PCA-N as the preferred method based on its origin in our laboratory, and we appreciate the reviewer's attention to potential bias in our manuscript.

Yes, PCA-N has notable limitations as it exhibits higher variability and is more susceptible to platelet contamination, which is harder to remove than erythrocyte-derived proteins. While PCA-N shows relative resistance to erythrocyte contamination (**Figure 1**), **Figures S3** and **S5** clearly demonstrate the continued influence of abundant erythrocyte proteins.

In response, we have revised the manuscript to remove language suggesting PCA-N is 'unique' or superior. We now present all workflows with a balanced discussion of their respective strengths and limitations. This includes explicitly noting PCA-N's greater variance, modest ID counts, and vulnerability to platelet contamination.

We agree that no single method should be positioned as best and thank the reviewer for helping us ensure a more objective and balanced comparison.

“Erythrocyte contamination, however, produced fundamentally different patterns across workflows. PCA-N demonstrated notable resistance, maintaining nearly identical protein numbers (~800-900) regardless of spiked-in erythrocyte concentration, while all other workflows suffered strongly from this.”

“Collectively, these results demonstrate that blood cell contamination has the potential to substantially impact protein identification in a workflow-dependent manner, with bead-based methods showing high susceptibility to cellular protein contaminations, while PCA-N shows particular resistance to erythrocyte contamination but remains susceptible to platelet contamination.”

“Bead-based workflows again showed high susceptibility to platelet contamination compared to the neat workflow whereas the PCA-N workflow showed reduced sensitivity to erythrocyte proteins.”

“These findings underscore our hypothesis that plasma contamination may distort proteome profiles in a workflow and cell-type specific manner. Bead-based methods demonstrate high susceptibility to cellular contamination, particularly from platelets and PBMCs, which can lead to dramatically inflated protein identification numbers that may not reflect the true circulating plasma proteome.”

Page 3: "Exploring the biological mechanism"

I was expecting a biological interpretation of why certain proteins are enriched or depleted under different conditions. The phrase "biological mechanism" suggested a functional or mechanistic explanation that I feel, I did not get.

The reviewer correctly points out that the term 'biological mechanism' implies a functional or mechanistic explanation, which we did not provide. Our intent was to describe differential protein detection patterns across workflows, not to investigate the biological or chemical mechanisms underlying those patterns.

We have revised the text to use more accurate terminology that reflects the scope of our analysis: a comparison of protein enrichment and depletion profiles across workflows under varying pre-analytical conditions.

“This involves experiments across multiple dimensions including pre-analytical variations commonly encountered in clinical studies, examining the differential protein detection patterns across workflows and analyzing their quantitative performance (Figure 1A-D).”

Given the workflow's sensitivity to small cellular contaminants, a statement clarifying how residual intact yeast cells were excluded would be useful.

To minimize contamination from residual intact yeast cells, we subjected the lysate to high-speed centrifugation at $16,000 \times g$ for 20 minutes following bead milling. This step pelleted unlysed cells and debris, and the supernatant was carefully collected without disturbing the pellet. We are confident this protocol ensures that only soluble proteins were carried forward for analysis. We have added this clarification to the manuscript:

“Soluble yeast proteins obtained through cell disruption and centrifugation were spiked into plasma at 8 different ratios for the 5 workflows in quadruplicates (1:2, 1:4, 1:10, $1:10^2$, $1:10^3$, $1:10^4$, $1:10^5$, $1:10^6$, and pure plasma) (Methods) (Figure 3A). To address the potential for residual intact yeast cells, the lysate was subjected to high-speed centrifugation at $16,000 \times g$ for 20 minutes after cell disruption. This rigorous centrifugation effectively pellets intact cells and large cellular debris, with careful supernatant collection to ensure that only soluble proteins were carried forward for subsequent analyses, minimizing contamination from residual intact yeast cells.”

Page 11: "Contribution of freeze-thawing was minimal"

While freeze-thawing may not alleviate the effect of cellular contamination it still had a strong effect on the variance in the data, especially when samples were not centrifuged. I would therefore recommend to rephrase this statement.

We agree: while freeze-thaw cycles do not reduce cellular contamination, they significantly increased variability in the data, especially in bead-based workflows. We have revised the manuscript to more accurately reflect this effect, acknowledging the observed increase in variance.

“Analysis of CVs revealed slight differences between freeze-thaw and direct processing conditions in several workflows, particularly in non-magnetic beads and SAX magnetic beads, although this effect was less pronounced in neat and PCA-N workflows (**Fig 5B**).”

“A Principal Component Analysis (PCA) pointed to platelet contamination as the dominant source of variation. While freeze-thaw cycles had less impact on overall proteome profiles compared to contamination, they notably increased data variability, particularly in bead-based workflows (**Figure 5C**).”

A recommendation on dealing with acquired or already published datasets would be useful. The authors present contamination indices but it is unclear how they can/should be used: excluding samples, forming experimental sub-groups, is correction possible?

We agree that clearer guidance is needed on how contamination indices can be applied, particularly in retrospective or already published datasets.

We recommend a three-step strategy for implementing contamination control, applicable to both prospective and retrospective studies:

1. Sample quality assessment: Calculate contamination indices and exclude outlier samples with excessive contamination relative to others in the same dataset. Since baseline contamination levels vary across studies and workflows, thresholds should be determined relative to the distribution within each dataset, rather than using fixed cutoffs.
2. Bias detection: Assess whether contamination levels differ systematically between groups (e.g., cases vs. controls), as this can confound biological interpretation.
3. Biomarker evaluation: Examine whether candidate biomarkers overlap with contamination marker panels or correlate with known contamination markers, these may represent technical artifacts rather than true biological signals.

This approach is now expanded on in the revised Discussion, and we have included the illustrative figure from our previous work (**Response Figure 7**, Geyer et al., 2019)

“To mitigate these risks, we had previously developed a three-step contamination control strategy that provides practical guidance for both prospective and retrospective studies: (1) assessing contamination in individual samples using our validated marker panels and excluding outliers relative to study-specific baselines, (2) detecting potential bias between study groups, and (3) correlating candidate biomarkers with cell-specific markers using global correlation maps to identify artifacts and distinguish genuine biological signals from technical effects.

Response Figure 7 - **Quality marker panels in a weight loss study and literature study**

(A) Assessment of individual sample quality with respect to the three contamination indices using the online tool at www.plasmaproteomeprofiling.org. Samples with indices that are more than two standard deviations from the mean (horizontal red lines) are flagged as potentially contaminated (red bars and sample numbers).

(B) Volcano plot of the proteome comparison of time point 1 vs. 4. Proteins of the platelet panel are highlighted in blue and two additional significantly regulated proteins in red.

(C) Global correlation map on the left with an inset of the platelet cluster on the right. The two significant outliers of the volcano plot in (B) are marked in red. Platelet panel proteins are highlighted in blue in the inset. Red patches in the global correlation map indicate positive and blue patches negative correlations.

(D) Literature analysis of 210 publications using MS-based plasma proteomics to identify new biomarkers. The number of quality markers reported as biomarker candidates in these studies is indicated.

(E) Distribution of the reported quality markers according to the three types of likely contaminations. The distribution is shown across studies that report one, two, or three proteins of the same quality marker panel.

Wishlist:

Enrichment/Depletion analysis of the samples after the 30-minute centrifugation - similar or in combination to Figure 1E. This could offer a cleaner view of which proteins are truly plasma-derived.

We thank the reviewer for this excellent suggestion. In response, we performed an enrichment/depletion analysis of samples after 30-minute centrifugation to help distinguish plasma-derived proteins from cellular contaminants (**Response Figure 8** and **Appendix Figure S9**).

This analysis ranks proteins by intensity in non-centrifuged samples and compares their abundance before and after centrifugation across workflows. The neat workflow shows minimal changes, with only subtle shifts in individual proteins, including platelet markers. PCA-N shows selective depletion of platelet proteins, but limited enrichment overall.

In contrast, bead-based workflows exhibit a dual effect: strong depletion of contaminating proteins, especially platelet markers, and simultaneous enrichment of low-abundance proteins. This is most pronounced in the non-magnetic bead workflow, with similar trends in SAX and Sera Sil 700. We hypothesize that removal of cellular proteins frees up bead binding capacity, allowing more low-abundance plasma proteins to be captured. This suggests that bead saturation may be a limiting factor, and that contamination removal not only improves sample purity but can enhance detection of genuine plasma constituents.

This analysis reinforces our conclusions about workflow-dependent contamination and highlights the interplay between contaminant removal and low-abundance protein enrichment in bead-based protocols.

“To provide a comprehensive view of protein-level changes during contamination removal, we performed enrichment/depletion analysis comparing protein intensities before and after centrifugation across all workflows (**Appendix Figure S9**). This analysis revealed workflow-dependent patterns where bead-based methods show simultaneous depletion of contaminating proteins and enrichment of low-abundance plasma constituents. This suggests that contamination removal may enhance detection of genuine plasma proteins by freeing up bead binding capacity.”

Response Figure 8 - **Enrichment/depletion analysis before and after 30-minute centrifugation across workflows**

(A) All proteins: Proteins ranked by intensity in non-centrifuged samples (black dots) with corresponding intensities after centrifugation shown as enriched (red lines) or depleted (blue lines) across all five workflows. Each row represents different conditions, each column different workflows.

(B) Top 100 platelet markers: Same analysis focused specifically on the top 100 platelet-specific proteins.

It seems the "pure" plasma used in the initial experiments may still contain residual cells. For example, in Figure 5B (pure plasma barplot) and Figure 5C (Sera Sil 700), there are hints of

platelet contamination, as post-centrifugation samples appear to shift toward the contaminated profile.

We thank the reviewer for this astute observation. Our 'pure' plasma samples indeed still contain detectable levels of platelet contamination, which decrease after centrifugation, as seen in Figures 5B and 5C. This highlights the difficulty of obtaining truly platelet-free plasma, even with rigorous protocols involving fast and prolonged centrifugation.

Rather than undermining our results, this observation reinforces the central message of this study: complete removal of platelets is practically unachievable, even under controlled conditions. This makes systematic quality assessment even more essential. If even carefully prepared samples show residual contamination detectable by sensitive workflows, routine clinical samples are likely to carry substantially more.

This finding underscores the need for the contamination indices and three-step evaluation strategy we propose, which can help distinguish technical artifacts from true biological signals in plasma proteomics.

14th Aug 2025

Dear Dr. Mann,

Thank you for the submission of your revised manuscript to EMBO Molecular Medicine and please accept my apologies for the delay in getting back to you due to the holiday season. I am pleased to inform you that we will be able to accept your manuscript pending the following final amendments:

- 1) Please implement referee #3 suggestion.
- 2) In the main manuscript file, please do the following:
 - Please correct the order and headings of the sections to: Abstract / Keywords / The Paper Explained / Introduction / Results / Discussion / Methods / Data Availability / Acknowledgements / Disclosure and Competing Interests Statement / References / Main Figure Legends / Expanded View Figure Legends.
 - Please rename the expanded view figures and callouts to "Figure EV1" etc.
 - Please add callouts for the individual EV Figure panels where available.
 - Please indicate in the legend of Figure 2 that panels F-H are reused in Appendix Figure 4A.
 - In Methods, provide the statement that informed consent was obtained from all human subjects and confirm that the experiments conformed to the principles set out in the WMA Declaration of Helsinki and the Department of Health and Human Services Belmont Report.
 - Rename "Competing interests" to "Disclosure and competing interests statement". We updated our journal's competing interests policy in January 2022 and request authors to consider both actual and perceived competing interests. Please review the policy <https://www.embopress.org/competing-interests> and update your competing interests if necessary.
 - Author contributions: Please remove it from the manuscript and specify author contributions in our submission system. CRediT has replaced the traditional author contributions section because it offers a systematic machine-readable author contributions format that allows for more effective research assessment. You are encouraged to use the free text boxes beneath each contributing author's name to add specific details on the author's contribution. More information is available in our guide to authors: <https://www.embopress.org/page/journal/17574684/authorguide#authorshipguidelines>
 - Please use the following format to report the accession number of your data:

The datasets produced in this study are available in the following databases:
[data type]: [full name of the resource] [accession number/identifier] ([doi or URL or identifiers.org/DATABASE:ACCESSION])

Please check "Author Guidelines" for more information.

<https://www.embopress.org/page/journal/17574684/authorguide#availabilityofpublishedmaterial>

- 3) Tables: Please rename the expanded view table to "Table EV1", remove the legend from the manuscript text and added it to the table. Please also add legends to all the dataset files, in a separate tab/worksheet, which contain the title ("Dataset EV1" etc.) and a short description.
- 4) Source data: Please upload the source data files as one (ZIP) file per figure and upload a completed source data checklist sent to you on June 5.
- 5) Funding: Please add all sources of funding to "Acknowledgments".
- 6) The Paper Explained: Please add it to the main manuscript text.
- 7) Synopsis:
 - Synopsis image: Please resize the image to 550 px-wide x 300-600 pixels high and upload it as a high-resolution jpeg file.
 - Please check your synopsis text and image before submission with your revised manuscript. Please be aware that in the proof stage minor corrections only are allowed (e.g., typos).
- 8) As part of the EMBO Publications transparent editorial process initiative (see our Editorial at <http://embomolmed.embopress.org/content/2/9/329>), EMBO Molecular Medicine will publish online a Review Process File (RPF) to accompany accepted manuscripts. This file will be published in conjunction with your paper and will include the anonymous referee reports, your point-by-point response and all pertinent correspondence relating to the manuscript. Let us know whether you agree with the publication of the RPF and as here, if you want to remove or not any figures from it prior to publication. Please note that the Authors checklist will be published at the end of the RPF.
- 9) Please provide a point-by-point letter INCLUDING my comments as well as the reviewer's reports and your detailed responses (as Word file).

I look forward to reading a new revised version of your manuscript as soon as possible.

Yours sincerely,

Zeljko Durdevic

Zeljko Durdevic
Senior Editor
EMBO Molecular Medicine

*** Instructions to submit your revised manuscript ***

To submit your manuscript, please follow this link:

<https://embomolmed.msubmit.net/cgi-bin/main.plex>

- 1) a .docx formatted version of the manuscript text (including Figure legends and tables)
 - 2) Separate figure files*
 - 3) supplemental information as Expanded View and/or Appendix. Please carefully check the authors guidelines for formatting Expanded view and Appendix figures and tables at <https://www.embopress.org/page/journal/17574684/authorguide#expandedview>
 - 4) a letter INCLUDING the reviewer's reports and your detailed responses to their comments (as Word file).
 - 5) The paper explained: EMBO Molecular Medicine articles are accompanied by a summary of the articles to emphasize the major findings in the paper and their medical implications for the non-specialist reader. Please provide a draft summary of your article highlighting
 - the medical issue you are addressing,
 - the results obtained and
 - their clinical impact.This may be edited to ensure that readers understand the significance and context of the research. Please refer to any of our published articles for an example.
 - 6) Author contributions: the contribution of every author must be detailed in a separate section.
 - 7) EMBO Molecular Medicine now requires a complete author checklist (<https://www.embopress.org/page/journal/17574684/authorguide>) to be submitted with all revised manuscripts. Please use the checklist as guideline for the sort of information we need WITHIN the manuscript. The checklist should only be filled with page numbers where the information can be found. This is particularly important for animal reporting, antibody dilutions (missing) and exact values and n that should be indicated instead of a range.
 - 8) Every published paper now includes a 'Synopsis' to further enhance discoverability. Synopses are displayed on the journal webpage and are freely accessible to all readers. They include a short stand first (maximum of 300 characters, including space) as well as 2-5 one sentence bullet points that summarise the paper. Please write the bullet points to summarise the key NEW findings. They should be designed to be complementary to the abstract - i.e. not repeat the same text. We encourage inclusion of key acronyms and quantitative information (maximum of 30 words / bullet point). Please use the passive voice. Please attach these in a separate file or send them by email, we will incorporate them accordingly.
- You are also welcome to suggest a striking image or visual abstract to illustrate your article. If you do please provide a jpeg file 550 px-wide x 300-600px high.
- 9) A Conflict of Interest statement should be provided in the main text

10) Please note that we now mandate that all corresponding authors list an ORCID digital identifier. This takes <90 seconds to complete. We encourage all authors to supply an ORCID identifier, which will be linked to their name for unambiguous name identification.

Currently, our records indicate that the ORCID for your account is 0000-0003-1292-4799.

Link Not Available

11) Include a Reagents and Tools Table as part of the Methods section, which can be downloaded from our author guidelines (<https://www.embopress.org/page/journal/17574684/authorguide#structuredmethods>)

Photos 400-800 DPI

*Additional important information regarding figures and illustrations can be found at <https://bit.ly/EMBOPressFigurePreparationGuideline>. See also figure legend preparation guidelines: <https://www.embopress.org/page/journal/17574684/authorguide#figureformat>

***** Reviewer's comments *****

Referee #2 (Comments on Novelty/Model System for Author):

I would like to reiterate my earlier comments regarding the strong technical quality of the work, its medical relevance and the adequacy of the chosen model system.

Referee #2 (Remarks for Author):

Thank you for addressing the comments and requests. Your revisions helped improve the overall quality of the paper and clarify potential misunderstandings.

Referee #3 (Remarks for Author):

I appreciate the authors' thorough and detailed responses to my comments. My concerns have been addressed to my satisfaction, and I thank the authors for their careful revisions.

After reading the concerns of the other reviewers, the statement regarding impure samples - "If even carefully prepared samples show residual contamination detectable by sensitive workflows, routine clinical samples are likely to carry substantially more" - and the recommendation to remove contaminated samples, I worry that the manuscript might create scepticism in the broader community toward MS-based plasma proteomics. While the low-abundant proteome is partially due to cellular contamination, this is not necessarily prohibitive for the discovery of new biomarkers or for gaining biomedical insight, and can itself be a source of new findings. Similar challenges are likely also present in affinity-based proteomic approaches. It might be helpful if the authors could include a brief statement in the discussion offering a more positive perspective, so as to balance the message for preanalytical consideration and highlight the potential for plasma proteomics despite these challenges.

Reviewer #1

-

Reviewer #2

Comments on Novelty/Model System for Author:

I would like to reiterate my earlier comments regarding the strong technical quality of the work, its medical relevance and the adequacy of the chosen model system.

Remarks for Author:

Thank you for addressing the comments and requests. Your revisions helped improve the overall quality of the paper and clarify potential misunderstandings.

Reviewer #3

Remarks for Author:

I appreciate the authors' thorough and detailed responses to my comments. My concerns have been addressed to my satisfaction, and I thank the authors for their careful revisions.

After reading the concerns of the other reviewers, the statement regarding impure samples - "If even carefully prepared samples show residual contamination detectable by sensitive workflows, routine clinical samples are likely to carry substantially more" - and the recommendation to remove contaminated samples, I worry that the manuscript might create scepticism in the broader community toward MS-based plasma proteomics. While the low-abundant proteome is partially due to cellular contamination, this is not necessarily prohibitive for the discovery of new biomarkers or for gaining biomedical insight, and can itself be a source of new findings. Similar challenges are likely also present in affinity-based proteomic approaches. It might be helpful if the authors could include a brief statement in the discussion offering a more positive perspective, so as to balance the message for preanalytical consideration and highlight the potential for plasma proteomics despite these challenges.

We agree that while contamination poses challenges, it should not overshadow the tremendous potential of plasma proteomics. In the revised Discussion, we have therefore added a balancing statement highlighting that with ongoing advances in MS-based technologies and quality control strategies, the field is increasingly able to counteract pre-analytical biases. We emphasize that these improvements enhance the likelihood of identifying true biomarker candidates and strengthen the translational value of plasma proteomics despite the challenges of contamination.

"With the ongoing advancements in instrumentation and novel strategies to increase the dynamic range of the plasma proteome, challenges such as sample contamination can still arise. However, the MS-based proteomics field now has powerful strategies to counter

these negative effects. These developments greatly increase the likelihood of discovering true biomarker candidates that are worthwhile to carry forward to clinical validation.”

28th Aug 2025

Dear Dr. Mann,

We are pleased to inform you that your manuscript is accepted for publication and is now being sent to our publisher to be included in the next available issue of EMBO Molecular Medicine.

Zeljko Durdevic
Senior Editor
EMBO Molecular Medicine
